# MVSFormer++: Revealing the Devil in Transformer's Details for Multi-View Stereo

**Chenjie Cao**[1,2*†], **Xinlin Ren**[2*], **Yanwei Fu**[1‡]
[1] School of Data Science, Fudan University, [2] Shanghai Key Lab of Intelligent
Information Processing, School of Computer Science, Fudan University
`{cjcao20,xlren20,yanweifu}@fudan.edu.cn`

## Abstract

Recent advancements in learning-based Multi-View Stereo (MVS) methods have prominently featured transformer-based models with attention mechanisms. However, existing approaches have not thoroughly investigated the profound influence of transformers on different MVS modules, resulting in limited depth estimation capabilities. In this paper, we introduce MVSFormer++, a method that prudently maximizes the inherent characteristics of attention to enhance various components of the MVS pipeline. Formally, our approach involves infusing cross-view information into the pre-trained DINOv2 model to facilitate MVS learning. Furthermore, we employ different attention mechanisms for the feature encoder and cost volume regularization, focusing on feature and spatial aggregations respectively. Additionally, we uncover that some design details would substantially impact the performance of transformer modules in MVS, including normalized 3D positional encoding, adaptive attention scaling, and the position of layer normalization. Comprehensive experiments on DTU, Tanks-and-Temples, BlendedMVS, and ETH3D validate the effectiveness of the proposed method. Notably, MVSFormer++ achieves state-of-the-art performance on the challenging DTU and Tanks-and-Temples benchmarks. Codes and models are available at `https://github.com/maybeLx/MVSFormerPlusPlus`.

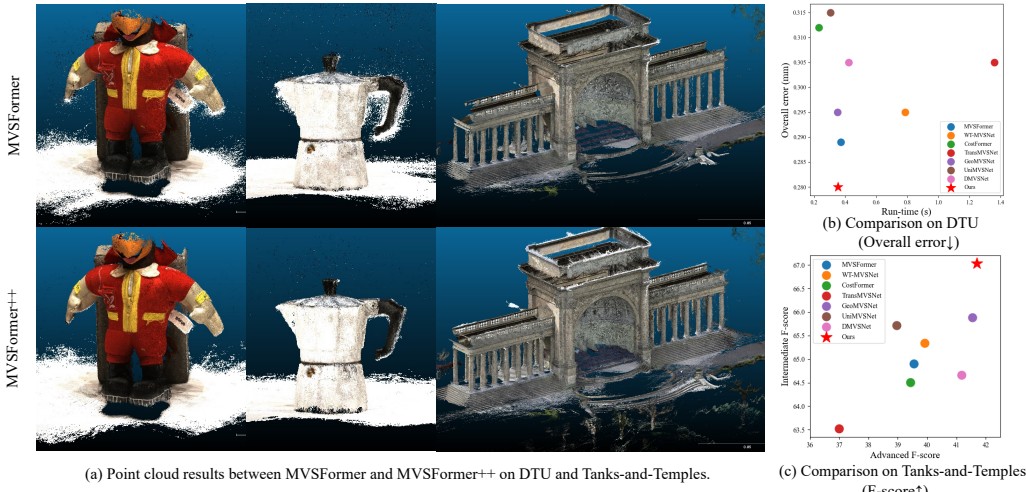

(a) Point cloud results between MVSFormer and MVSFormer++ on DTU and Tanks-and-Temples.

(b) Comparison on DTU (Overall error↓)

(c) Comparison on Tanks-and-Temples (F-score↑)

Figure 1: (a) Point cloud results compared between MVSFormer (Cao et al., 2022) and the proposed MVSFormer++ on DTU and Tanks-and-Temples. Results of state-of-the-art MVS methods on (b) DTU and (c) Tanks-and-Temples benchmark.

---

*Equal Contributions.

†Dr. Cao is at Damo, Alibaba Group now. This work was accomplished while he was at the School of Data Science, Fudan University.

‡Corresponding Author.

# 1 INTRODUCTION

Multi-View Stereo (MVS) is a crucial task in the field of Computer Vision (CV) with the objective of recovering highly detailed and dense 3D geometric information from a collection of calibrated images. MVS primarily involves the extraction of robust features and the precise establishment of correspondences between reference views and source views along epipolar lines. In contrast to traditional methods that address instance-level MVS problems through iterative propagation and matching processes, as in Furukawa & Ponce (2009); Galliani et al. (2015b); Schönberger et al. (2016), recent learning-based approaches have emerged as compelling alternatives. These innovative MVS approaches, such as Yao et al. (2018); Gu et al. (2020); Giang et al. (2022); Peng et al. (2022); Zhang et al. (2023c), have shown the potential to achieve remarkable reconstruction quality through end-to-end pipelines, particularly in complex scenes.

The essence of MVS learning can be fundamentally grasped as a feature-matching task conducted along epipolar lines, assuming known camera poses (Ding et al., 2022). Recent studies have underscored the importance of incorporating long-range attention mechanisms in various matching-based tasks, including image matching (Sun et al., 2021; Tang et al., 2022; Chen et al., 2022), optical flow (Huang et al., 2022; Shi et al., 2023a; Dong et al., 2023), and stereo matching (Li et al., 2021). Moreover, transformer modules (Vaswani et al., 2017; Yu et al., 2022) significantly improved the capacity for attention, leading to their widespread adoption in CV tasks, as exemplified by Vision Transformers (ViTs) (Dosovitskiy et al., 2020; He et al., 2022; Oquab et al., 2023). Consequently, the integration of transformer into MVS learning becomes a burgeoning research area (Ding et al., 2022; Liao et al., 2022; Chen et al., 2023). Among these methods, the pioneering work of MVSFormer (Cao et al., 2022) stands out for unifying pre-trained ViTs for feature extraction with integrated architectures and training strategies, resulting in advancements in the state-of-the-art of MVS[1].

While transformer-based MVS approaches have made significant strides, several unaddressed challenges remain, offering opportunities for further integration of transformers and MVS learning. 1) *Tailored attention mechanisms for different MVS modules*. Within the MVS learning framework, there exist two primary components: the feature encoder and cost volume regularization. These modules should not rely on identical attention mechanisms due to their distinct feature properties. 2) *Incorporating cross-view information into Pre-trained ViTs*. Despite the substantial improvements that pre-trained ViTs offer in MVSFormer, there remains a need for essential feature interaction across different views. Existing cross-view pre-trained ViTs have struggled to fully address the indispensable multi-view correlations. 3) *Enhancing Transformer's Length Extrapolation Capability in MVS*. A noticeable disparity exists between the image sizes during training and testing phases in MVS. Notably, feature matching at higher resolutions often leads to superior precision. Nevertheless, enabling transformers to generalize effectively to diverse sequential lengths, akin to Convolutional Neural Networks (CNNs), poses a substantial challenge.

We have conducted an exhaustive investigation into the transformer design, building upon the foundation of MVSFormer to address the aforementioned challenges. The approach has resulted in an enhanced iteration known as MVSFormer++. We begin by summarizing notable transformer-based MVS methods in Tab. 1, highlighting the innovations introduced by MVSFormer++. In particular, we have integrated the pre-trained DINOv2 (Oquab et al., 2023) as our powerful feature encoder. To improve the cross-view learning ability of DINOv2, we incorporate the meticulously designed Side View Attention (SVA) aside to DINOv2 layers, which incrementally injects cross-view attention modules employing linear attention mechanisms (Katharopoulos et al., 2020). Notably, our findings reveal that linear attention, based on feature aggregation, performs exceptionally well across various image sizes during the feature encoding stage, surpassing other attention mechanisms. Furthermore, we have identified the subsidiary but critical roles played by normalized 2D Positional Encoding (PE), Adaptive Layer Scaling (ALS), and the order of Layer Normalization (LN) in feature extraction, ensuring stable convergence and generalization.

Regarding cost volume regularization, employing linear attention to aggregate the cost volume along feature channels performs unsatisfactorily. This stems from the inherent characteristics of features within the cost volume, which heavily rely on group-wise feature dot products and variances, resulting in fewer feature-level representations. In contrast, vanilla attention excels in aggregating features

---

[1] MVSFormer has consistently held the high ranking on the Tanks-and-Temples intermediate benchmark (Knapitsch et al., 2017) since May 2022.

Table 1: Comparison of transformer-based MVS methods, including TransMVSNet (Ding et al., 2022), WT-MVSNet (Liao et al., 2022), CostFormer (Chen et al., 2023), and MVSFormer (Cao et al., 2022). MVSFormer++ surpasses other competitors with a meticulously designed transformer architecture, including attention with global receptive fields, transformer learning for both feature encoder and cost volume, cross-view attention, adaptive scaling for different sequence lengths, and specifically proposed positional encoding for MVS.

| Methods | Attention | Transformers work in | | Cross-view | Adaptive scaling | Positional Encoding (PE) | | |
|---|---|---|---|---|---|---|---|---|
| | global/window | Feature encoder | Cost volume | | | Abs./Rel. | Normalized | 3D-PE |
| TransMVSNet | global | ✓ | ✗ | ✓ | ✗ | absolute | ✗ | ✗ |
| WT-MVSNet | window | ✓ | ✓ | ✓ | ✗ | relative | ✗ | ✗ |
| CostFormer | window | ✗ | ✓ | ✗ | ✗ | relative | ✗ | ✗ |
| MVSFormer | global | ✓ | ✗ | ✗ | ✗ | absolute | ✓ | ✗ |
| MVSFormer++ | global | ✓ | ✓ | ✓ | ✓ | absolute | ✓ | ✓ |

along spatial dimensions, making it better suited for denoising the cost volume. However, integrating vanilla attention into an extensive 3D sequence presents significant challenges. While computational constraints can be alleviated through efficient attention implementations (Dao, 2023), vanilla attention still suffers from limited length extrapolation and attention dilution. To this end, we propose an innovative solution in the form of 3D Frustoconical Positional Encoding (FPE). FPE provides globally normalized 3D positional cues, enhancing the capacity to process diverse 3D sequences of extended length. Furthermore, we revisit the role of attention scaling and re-propose Adaptive Attention Scaling (AAS) to mitigate attention dilution. Our proposed Cost Volume Transformer (CVT) has proven to be remarkably effective with a simple design. It substantially elevates the final reconstruction quality, notably reducing the number of outliers, as depicted in Fig. 1(a).

In summary, our contributions can be highlighted as follows: 1) *Customized attention mechanisms*: We analyzed the components of MVS and strategically assigned distinct attention mechanisms based on their unique feature characteristics for different components. The tailored mechanism improves the performance of each component for the MVS processing. 2) *Introducing SVA*, a novel approach to progressively integrating cross-view information into the pre-trained DINOv2. This innovation significantly strengthens depth estimation accuracy, resulting in substantially improved MVS results based on pre-trained ViTs. 3) *In-depth transformer design*: Our research delves deep into the intricacies of transformer module design. We present novel elements like 2D and 3D-based PE and AAS. These innovations address challenges of length extrapolation and attention dilution. 4) *Setting new performance standards*: MVSFormer++ attains state-of-the-art results across multiple benchmark datasets, including DTU, Tanks-and-Temples, BlendedMVS, and ETH3D. Our model's outstanding performance demonstrates effectiveness and competitiveness in the field of MVS research.

## 2 RELATED WORKS

**Learning-based MVS Methods** (Yao et al., 2018; Gu et al., 2020; Wang et al., 2021a; Peng et al., 2022) strengthened by Deep Neural Networks (DNNs) have achieved prominent improvements recently. Yao et al. (2018) proposed an end-to-end network MVSNet to address the MVS issue through three key stages, including feature extraction, cost volume formulation, and regularization. A 3D CNN is further used to regress the depth map. Yao et al. (2019); Yan et al. (2020); Wei et al. (2021); Cai et al. (2023); Xu et al. (2023) iteratively estimate depth residuals to overcome the heavy computation from 3DCNN learned for the cost volume regularization. On the other hand, coarse-to-fine learning strategies (Gu et al., 2020; Cheng et al., 2020; Mi et al., 2022) are proposed to refine the multi-scale depth maps, enjoying both proper performance and efficient memory cost, which is widely used in the MVS pipeline. Moreover, many researchers try to learn reliable cost volume formulation according to the visibility of each view (Zhang et al., 2020b; Wang et al., 2022) and adaptive depth ranges (Li et al., 2023; Zhang et al., 2023a; Cheng et al., 2020). Besides, auxiliary losses based on SDF (Zhang et al., 2023b), monocular depth (Wang et al., 2022), and neural rendering (Xi et al., 2022; Shi et al., 2023b) also improve the MVS performance.

**Transformers for Feature Correlation.** Due to the ability to capture long-range contextual information, transformers are widely used in feature matching (Sun et al., 2021; Tang et al., 2022; Cao & Fu, 2023), optical flow (Dong et al., 2023), and stereo matching (Li et al., 2021). These manners stack self and cross-attention blocks to learn feature correlations between two frames. Inspired by them, TransMVSNet (Ding et al., 2022) incorporate feature matching transformers into the MVS to aggregate features between source and reference views. Other works like Liao et al. (2022); Chen

et al. (2023) apply shifted-window attention along the epipolar lines and cost volumes to improve the performance. However, note that many nuanced facets of transformer models, especially in the context of low-level tasks like MVS, remain relatively underexplored in comprehensive research.

**Attention in Transformers.** The conventional attention mechanism (Vaswani et al., 2017) aggregates value features based on the correlation of queries and keys. However, this vanilla attention scheme is notorious for its quadratic computational complexity. Fortunately, recent pioneering advancements in IO-Awareness optimizations largely alleviate this problem by FlashAttention (Dao et al., 2022; Dao, 2023). Consequently, vanilla attention remains the prevailing choice for large-scale models CV (Rombach et al., 2022) and NLP (Brown et al., 2020; Touvron et al., 2023). In pursuit of enhanced efficiency, ViTs have embarked on an exploration of diverse attention mechanisms. These include linear attention (Katharopoulos et al., 2020; Shen et al., 2021), shifted window attention (Liu et al., 2021), Pyramid ViT (PVT) (Wang et al., 2021b; Chu et al., 2021), and Top-K attention (Tang et al., 2022; Zhu et al., 2023). It is important to note that all these attention variants exhibit unique strengths and weaknesses, which are primarily discussed within the context of classification and segmentation tasks, rather than in the realm of low-level tasks.

**LN and PE in Transformers.** Dong et al. (2021) have demonstrated that utilizing pure attention modules can lead to a phenomenon known as rank collapse. In practice, a complete transformer block typically consists of not only the attention module but also skip connections, a Feed Forward Network (FFN), and Layer Normalization (LN), as discussed in Metaformer (Yu et al., 2022). Notably, while LN alone may not effectively prevent rank collapse, its various usage patterns play pivotal roles in balancing the trade-off between model convergence and generalization (Dong et al., 2021). Specifically, Post-LN (Vaswani et al., 2017) tends to enhance generalization, whereas Pre-LN, as explored in Wang et al. (2019), offers greater stability during convergence. Additionally, the concept of zero-initialized learnable residual connections in Bachlechner et al. (2021) can be employed as an alternative to LN within the transformer architecture. Furthermore, PE serves a crucial role in providing positional information to unordered sequences processed by transformers. These encodings come in various forms, including absolute (Vaswani et al., 2017), relative (Raffel et al., 2020), and rotary (Su et al., 2021) positional encodings, each offering unique advantages. While convolutional layers with padding can implicitly capture the distance from the image boundary (Islam et al., 2020), our experimental findings corroborate that PE significantly influences the performance of MVS tasks.

## 3  REVEALING THE DEVIL IN TRANSFORMERS FOR MVS

**Preliminary.** MVSFormer (Cao et al., 2022) introduces a pioneering approach by harnessing pre-trained ViTs to enhance the learning process for MVS. It capitalizes on the synergies between features extracted from pre-trained ViTs and those obtained through the Feature Pyramid Network (FPN). This unique combination proves invaluable for effectively modeling reflective and texture-less regions. Furthermore, MVSFormer addresses the challenge posed by varying image resolutions between training and testing data through the implementation of a multi-scale training strategy. In addition to this, MVSFormer leverages the strengths of both regression and classification techniques for depth estimation. It optimizes the model using a cross-entropy loss while incorporating a temperature-based depth expectation mechanism for predicting depth during inference. This holistic approach enhances the accuracy and robustness of depth estimation.

Building upon the MVSFormer, MVSFormer++ adopts the latest DINOv2 (Oquab et al., 2023) as the frozen ViT backbone, which enjoys prominent zero-shot cross-domain feature matching ability. We verify the efficacy of DINOv2 on MVS in the pilot study (Tab. 8). More details

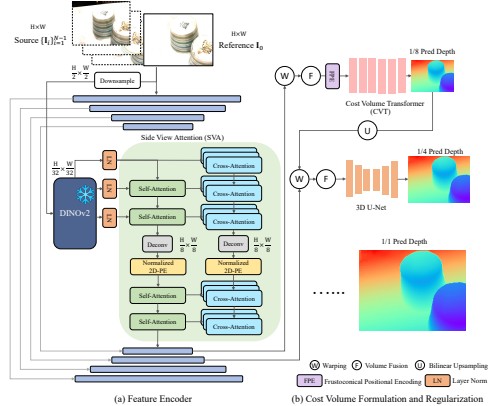

Figure 2: The Overview of MVSFormer++. (a) Feature extraction enhanced with SVA module, normalized 2D-PE, and Norm&ALS. (b) Multi-scale cost volume formation and regularization, where CVT is strengthed by FPE and AAS resulting in solid depth estimation.

about DINOv2 are discussed in Appendix A.1. Moreover, MVSFormer++ takes transformer-based learning a step further. It enhances both the feature encoder and cost volume regularization, consolidating the strengths of transformers in MVS for improved performance and versatility.

**Overview.** The overview of MVSFormer++ is presented in Fig. 2. Given $N$ calibrated images containing a reference image $\mathbf{I}_0$, and source view images $\{\mathbf{I}_i\}_{i=1}^{N-1}$, MVSFormer++ operates as a cascade MVS model, producing depth estimations that span from 1/8 to 1/1 of the original image size. Specifically, for the feature extraction, we employ FPN to extract multi-scale features $\{\hat{F}_i\}_{i=0}^{N-1}$. Subsequently, both the reference and source view images are downsampled by half and fed into the frozen DINOv2-base to extract high-quality visual features. To enrich the DINOv2 model with cross-view information, we propose Side View Attention (SVA) mechanism, enhanced with normalized 2D PE and adaptive layer scaling (Sec. 3.1). For the cost volume regularization, we apply the Cost Volume Transformer (CVT) strengthened by Frustoconical Positional Encoding (FPE) and Adaptive Attention Scaling module (AAS) to achieve solid depth initialization in the 1/8 coarse stage (Sec. 3.2).

## 3.1 TRANSFORMERS FOR FEATURE ENCODER

**Side View Attention (SVA).** To effectively capture extensive global contextual information across features from different views, we leverage SVA to further enhance the multi-layer DINOv2 features, denoted as $\{F_i^l\}_{i=0}^{N-1}$. SVA functions as a side-tuning module (Zhang et al., 2020a), *i.e.*, it can be independently trained without any gradients passing through the frozen DINOv2.

To learn cross-view information through attention modules, the interlaced self and cross-view attentions (Ding et al., 2022) are primarily beneficial for source features $\{F_i^l\}_{i=1}^{N-1}$, which learn to aggregate reference ones for better feature representations. In contrast, reference features just need to be encoded by self-attention modules. Our further investigation revealed that self-attention modules are unnecessary for source features from DINOv2. Thus, in SVA for DINOv2, we separately encode features of reference and source views by self and cross-attention as depicted in Fig. 3, which saves half of the computation without obvious performance degradation. Subsequently, after the

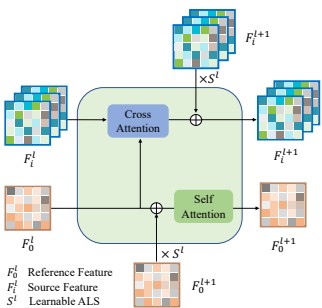

Figure 3: Illustration of SVA. Self and cross-view attention are separately used to learn reference and source features respectively.

feature aggregation from cross-attention, we add new DINOv2 features from level $l + 1$ with adaptive layer scaling to the next SVA block's inputs. As shown in Fig. 2, after the upsampling to the 1/8 scale, we further incorporate two additional SVA blocks to high-resolution features with normalized 2D-PE. Note that we systematically validate several attention mechanisms for SVA. Remarkably, linear attention (Katharopoulos et al., 2020) outperforms others in Tab. 5(b). This underscores the efficacy of linear attention when coupled with feature-level aggregation for DINOv2 features. Moreover, linear attention's inherent robustness allows it to gracefully accommodate diverse sequence lengths, effectively overcoming the limitations associated with vanilla attention in the context of MVS.

**Normalized 2D Positional Encoding (PE).** While DINOv2 already includes positional encodings (PE) for features at the 1/32 scale, we have taken it upon ourselves to further enrich the positional cues tailored for SVA. This enhancement facilitates the learning of high-resolution features at the 1/8 scale, as depicted in Fig. 2(a). In alignment with the principles in Chen et al. (2022), we have implemented a linear normalizing approach to ensure that the testing maximum values of height and width positions are equal to a consistent scale used in the training phase, specifically set at (128, 128). Such simple yet effective normalized 2D-PE has demonstrated its remarkable ability to yield robust depth estimation results when subjected to high-resolution image testing, as empirically validated in our preliminary investigation in Tab. 7 and Tab. 11.

**Normalization and Adaptive Layer Scaling (Norm&ALS).** In response to the substantial variance observed in DINOv2 multi-layer features (Fig. 6), we apply the LNs to normalize all the DINOv2 features before the SVA module. Moreover, all SVA blocks are based on the Pre-LN (Wang et al., 2019), which normalizes features before the attention and FFN blocks rather than after the residual addition as Post-LN. Pre-LN enjoys more significant gradient updates, especially when being trained

for multi-layer attention blocks (Wang et al., 2019). In Tab. 6, Pre-LN achieves superior performance, while we find that Post-LN usually struggles for slower convergence. Furthermore, we introduce the learnable ALS multiplied to normalized DINOv2 features, which adaptively adjust the significance of features from unstable frozen DINOv2 layers. The combination of Norm&ALS significantly enhances the training stability and convergence when stacking multi-layer transformer blocks, as shown in Fig. 7. Notably, the learnable coefficients $S^l$ are all initialized with 0.5 within MVSFormer++, emphasizing the impact for the latter layers as empirically verified in DINOv2 (Oquab et al., 2023).

**SVA vs Intra, Inter-Attention.** Despite some similarities in using self and cross-attention, our SVA differs from Intra, Inter-attention (Ding et al., 2022) in both purpose and implementation. The most critical difference is that SVA performs cross-view learning for both DINOv2 (1/32) and coarse MVS (1/8) features (Fig. 2), while Intra, Inter-attention only considers coarse MVS features. For DINOv2 features, SVA is specifically designed as a side-tuning module without gradient propagation through the frozen DINOv2, which efficiently incorporates cross-view information to monocular pre-trained ViT. ALS is further proposed to adaptively learn the importance of various DINOv2 layers, while Pre-LN is adopted to improve the training convergence. For coarse MVS features, we emphasize that normalized 2D-PE improves the generalization in high-resolution MVS. We also omit self-attention for features from DINOv2 source views to simplify the model with competitive performance.

## 3.2 TRANSFORMERS FOR COST VOLUME REGULARIZATION

In this section, we first analyze the feasibility of using transformer modules in the cost volume regularization. Then, we introduce how to tackle the limited length extrapolation capability for 3D features and the attention dilution issue with FPE and AAS respectively.

**Could Pure Transformer Blocks (CVT) Outperforms 3DCNN?** The cost volume regularization works as a denoiser to filter noisy feature correlations from the encoder. Most previous works leverage 3DCNN to denoise such cost volume features (Yao et al., 2018), while some transformer-based manners (Liao et al., 2022; Chen et al., 2023) are also built upon locally window-based attention. Contrarily, in this work, we embark on a comprehensive investigation that regards the whole noisy cost volume as a global sequential feature and then processes it through the pure transformer based on vanilla attention. Specifically, we first downsample the 4D group-wise cost volume correlation (Xu & Tao, 2020) through one layer of non-overlapping patch-wise convolution with stride $[2, 4, 4]$ to $\hat{C} \in \mathbb{R}^{C \times D \times H \times W}$, where $D, H, W$ indicate the dimension along depth, height, and width; $C = 64$ is the cost volume channel. Then the cost volume feature is rearranged to the shape of $(C \times DHW)$, while $DHW$ can be seen as the global sequence learned by transformer blocks. Thanks to the efficient FlashAttention (Dao, 2023), CVT eliminates the quadratic complexity of the vanilla attention as in Fig. 1(b). We stack 6-layer standard Post-LN-based transformer blocks with competitive computation compared to 3DCNN. Finally, output features are upsampled with another non-overlapping transposed convolution layer to the original size for achieving depth logits along all hypotheses.

We should clarify some details about CVT. First, we found that linear attention performs very poorly in CVT as in Tab. 5, which indicates that feature-level aggregation is unsuitable for learning correlation features after the dot product. Moreover, we only apply the CVT in the first coarse stage, while using CVT in other fine-grained stages would cause obviously inferior performance. In the cascade model, only the first stage enjoys a complete and continuous 3D scene, while all pixels within the same $d_i$ in $D$ share the same depth hypothesis plane. Therefore, we think that the integrality and continuity of the cost volume are key factors in unlocking the capacity of CVT.

**Frustoconical Positional Encoding (FPE).** To enhance the model's ability to generalize across a variety of image resolutions, we first normalize the 3D position $P \in \mathbb{R}^{3 \times DHW}$ of the cost volume into the range $[0, 1]^3$ through the frustum-shaped space built upon the nearest and farthest depth planes pre-defined for each scene as shown in Fig. 4(a). Then we separately apply 1D sinusoidal PE along the x, y, z dimensions, and encode them into $C$ channels for each axis. Subsequently, we concatenate all these three PE dimensions into FPE shaped as $(3C \times DHW)$, and apply a $1 \times 1$ convolutional layer to project them to the same channel of cost volume feature as $(C \times DHW)$. This projected FPE is then added to the cost volume feature. FPE helps the model in capturing both absolute and relative positions in 3D scenes, which is crucial for improving CVT's depth estimation. Note that FPE is only applied to the first stage's cost volume for CVT, while the zero padding-based 3DCNN has already captured sufficient positional clues for other stages (Islam et al., 2020).

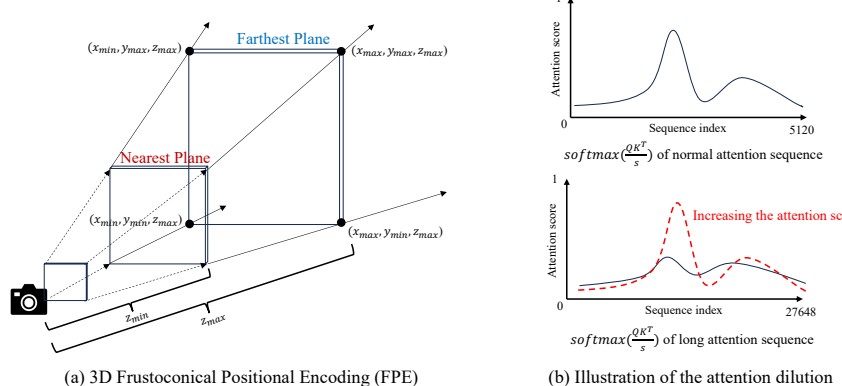

(a) 3D Frustoconical Positional Encoding (FPE)  (b) Illustration of the attention dilution

Figure 4: Illustration of 3D FPE and attention dilution. (a) We normalize all points in the cost volume within the nearest and farthest depth plane. (b) The attention score would be diluted when the sequence increases, making it challenging to correctly focus on related target values.

**Adaptive Attention Scaling (AAS).** We should clarify that FPE is insufficient to make CVT generalize to various cost volume lengths. As shown in Fig. 4(b), we analyze the phenomenon called attention dilution that primarily occurred in NLP (Chiang & Cholak, 2022). In CVT, the training sequential lengths of cost volume are around 6,000, while testing lengths are remarkably increased, varying from 27,648 to 32,640. Therefore, the attention score after the softmax operation is obviously diluted, which makes aggregated features fail to focus on correct target values. So attention dilution would hinder the performance of CVT for MVS with high-resolution images. One trivial solution is to train CVT on high-resolution images directly, but it would cause prohibitive computation and still lack generalization for larger test images. Su (2021) provided a perspective that we should keep the invariant entropy for the attention score as:

$$\mathcal{H}_i = -\sum_j^n a_{i,j} \log a_{i,j}, \qquad a_{i,j} = \frac{e^{\lambda q_i \cdot k_j}}{\sum_j^n e^{\lambda q_i \cdot k_j}}, \tag{1}$$

where $q_i, k_j$ is query and key features; $a_{i,j}$ is the attention score of query $i$ and key $j$; $\mathcal{H}_i$ is the entropy of query $i$; $n$ is the sequential length; $\lambda$ is the attention scaling. To make $\mathcal{H}_i$ independent of $n$, we could achieve $\lambda = \frac{\kappa \log n}{d}$ as proven by Su (2021), where $\kappa$ is a constant. Thus we could formulate the attention as:

$$\text{Attenion}(\mathbf{Q}, \mathbf{K}, \mathbf{V}) = \text{Softmax}(\frac{\kappa \log n}{d} \mathbf{Q} \mathbf{K}^T) \mathbf{V}, \tag{2}$$

where $d$ is the feature channel. Note that the default attention scale is $\lambda = \frac{1}{\sqrt{d}}$. We empirically set $\kappa = \frac{\sqrt{d}}{\log \overline{n}}$, where $\overline{n}$ is the mean sequential length of features during the multi-scale training (Cao et al., 2022). Thus, the training of CVT enhanced by AAS approaches to the normal transformer training with default attention scaling, while it could adaptively adjust the scaling for various sequential lengths to retain the invariant entropy during the inference as shown in Fig. 4(b). The newly repurposed AAS enjoys good generalization as verified in Sec. 4.1 even with 2k images from Tanks-and-Temples.

## 4 EXPERIMENT

**Implementation Details.** We train and test our MVSFormer++ on the DTU dataset (Aanæs et al., 2016) with five-view images as input. Following MVSFormer (Cao et al., 2022), our network applies 4 coarse-to-fine stages of 32-16-8-4 inverse depth hypotheses. We adopt the same multi-scale training strategy with the resolution scaling from 512 to 1280. Since the DTU dataset primarily consists of indoor objects with identical camera poses, in order to enhance the model's generalization capability for outdoor scenes such as Tanks-and-Temples (Knapitsch et al., 2017) and ETH3D dataset (Schops et al., 2017), we perform fine-tuning on a mixed dataset that combines DTU and BlendedMVS (Yao et al., 2020). Specifically, we train MVSFormer++ using Adam for 10 epochs at a learning rate of 1e-3 on the DTU dataset. Then we perform further fine-tuning of MVSFormer++ for 10 additional epochs with a reduced learning rate of 2e-4 on the mixed DTU and BlendedMVS dataset.

Table 2: Quantitative point cloud results (mm) on DTU (lower is better). The best results are in bold, and the second ones are underlined. *All scenes share the same threshold for the post-processing.*

| Methods | Accuracy↓ | Completeness ↓ | Overall↓ |
|---|---|---|---|
| Gipuma (Galliani et al., 2015a) | **0.283** | 0.873 | 0.578 |
| COLMAP (Schönberger et al., 2016) | 0.400 | 0.664 | 0.532 |
| CasMVSNet (Gu et al., 2020) | 0.325 | 0.385 | 0.355 |
| AA-RMVSNet (Wei et al., 2021) | 0.376 | 0.339 | 0.357 |
| UniMVSNet (Peng et al., 2022) | 0.352 | 0.278 | 0.315 |
| TransMVSNet (Ding et al., 2022) | 0.321 | 0.289 | 0.305 |
| WT-MVSNet (Liao et al., 2022) | 0.309 | 0.281 | 0.295 |
| CostFormer (Chen et al., 2023) | 0.301 | 0.322 | 0.312 |
| RA-MVSNet (Zhang et al., 2023b) | 0.326 | 0.268 | 0.297 |
| GeoMVSNet (Zhang et al., 2023c) | 0.331 | 0.259 | 0.295 |
| MVSFormer (Cao et al., 2022) | 0.327 | **0.251** | 0.289 |
| MVSFormer++ (ours) | 0.3090 | 0.2521 | **0.2805** |

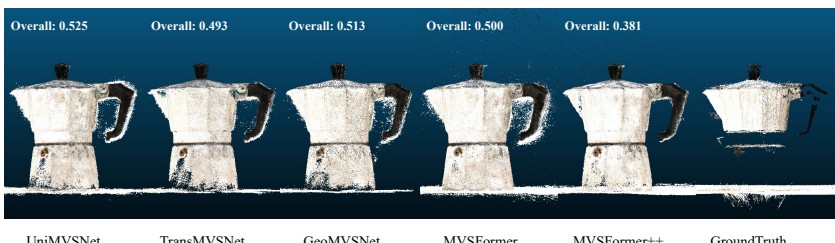

Figure 5: Qualitative results compared with state-of-the-art models on scan77 in DTU.

## 4.1 EXPERIMENTAL PERFORMANCE

**Evaluation on DTU Dataset.** We first resize the test image to $1152 \times 1536$ and set view number $N$ to 5. Then we use off-the-shelf Gipuma (Galliani et al., 2016) fusing depth maps to generate dense 3D point clouds for all the scans with identical hyper-parameters. The final results are evaluated on the official metrics as accuracy, completeness, and overall errors. In Tab. 2 we can observe that MVSFormer++ outperforms both traditional methods and learning-based methods at overall error by a large margin. As the extension of MVSFormer, our MVSFormer++ achieves more accurate and fewer outliers results as shown in Fig. 1(a) and Fig. 5.

**Evaluation on Tanks-and-Temples and ETH3D.** To evaluate the efficacy in terms of generalization to outdoor scenes, we evaluate our MVSFormer++ on Tanks-and-Temples online benchmark with 2k image sizes. Quantitative results are shown in Tab. 3, which showcases that our MVSFormer++ surpasses all other state-of-the-art methods with mean F-scores of 67.03 and 41.70 on the Intermediate and Advanced sets, respectively. More qualitative results are shown in Fig. 13 of the Appendix, which denotes the good generalization and impressive performance of MVSFormer++. Note that our AAS and FPE perform well under extremely high-resolution images. More results and discussions about ETH3D (Schops et al., 2017) are listed in Tab. 12 of Appendix Sec. A.4.

Table 3: Quantitative results of F-score on Tanks-and-Temples. A higher F-score means a better reconstruction quality. The best results are in bold, while the second ones are underlined.

| Methods | Intermediate | | | | | | | | | Advanced | | | | | | |
|---|---|---|---|---|---|---|---|---|---|---|---|---|---|---|---|---|
| | Mean | Fam. | Fra. | Hor. | Lig. | M60 | Pan. | Pla. | Tra. | Mean | Aud. | Bal. | Cou. | Mus. | Pal. | Tem. |
| COLMAP (Schönberger et al., 2016) | 42.14 | 50.41 | 22.25 | 26.63 | 56.43 | 44.83 | 46.97 | 48.53 | 42.04 | 27.24 | 16.02 | 25.23 | 34.70 | 41.51 | 18.05 | 27.94 |
| CasMVSNet (Gu et al., 2020) | 56.84 | 76.37 | 58.45 | 46.26 | 55.81 | 56.11 | 54.06 | 58.18 | 49.51 | 31.12 | 19.81 | 38.46 | 29.10 | 43.87 | 27.36 | 28.11 |
| CostFormer (Chen et al., 2023) | 64.51 | 81.31 | 65.65 | 55.57 | 63.46 | 66.24 | 65.39 | 61.27 | 57.30 | 39.43 | 29.18 | 45.21 | 39.88 | 53.38 | 34.07 | 34.87 |
| TransMVSNet (Ding et al., 2022) | 63.52 | 80.92 | 65.83 | 56.94 | 62.54 | 63.06 | 60.00 | 60.20 | 58.67 | 37.00 | 24.84 | 44.59 | 34.77 | 46.49 | 34.69 | 36.62 |
| WT-MVSNet (Liao et al., 2022) | 65.34 | 81.87 | 67.33 | 57.76 | 64.77 | 65.68 | 64.61 | 62.35 | 58.38 | 39.91 | 29.20 | 44.48 | 39.55 | 53.49 | 34.57 | 38.15 |
| RA-MVSNet (Zhang et al., 2023b) | 65.72 | 82.44 | 66.61 | 58.40 | 64.78 | **67.14** | 65.60 | **62.74** | 58.08 | 39.93 | 29.17 | 46.05 | 40.23 | 53.22 | 34.62 | 36.30 |
| D-MVSNet (Ye et al., 2023) | 64.66 | 81.27 | 67.54 | 59.10 | 63.12 | 64.64 | 64.80 | 59.83 | 56.97 | 41.17 | 30.08 | 46.10 | **40.65** | 53.53 | 35.08 | 41.60 |
| MVSFormer (Cao et al., 2022) | 66.37 | 82.06 | **69.34** | 60.49 | 68.61 | 65.67 | 64.08 | 61.23 | 59.53 | 40.87 | 28.22 | **46.75** | 39.30 | 52.88 | 35.16 | 42.95 |
| MVSFormer++ (ours) | **67.03** | **82.87** | 68.90 | **64.21** | **68.65** | 65.00 | **66.43** | 60.07 | **60.12** | **41.70** | **30.39** | 45.85 | 39.35 | **53.62** | **35.34** | **45.66** |

Table 4: Ablation results with different components on DTU test dataset, Metrics are depth error ratios of 2mm ($e_2$), 4mm ($e_4$), 8mm ($e_8$).

| CVT | FPE | AAS | SVA | Norm&ALS | $e_2 \downarrow$ | $e_4 \downarrow$ | $e_8 \downarrow$ | Accuracy$\downarrow$ | Completeness$\downarrow$ | Overall$\downarrow$ |
|-----|-----|-----|-----|----------|------|------|------|----------|--------------|---------|
| | | | | | 16.38 | 11.16 | 7.79 | 0.3198 | 0.2549 | 0.2875 |
| ✓ | | | | | 17.95 | 12.93 | 9.27 | 0.3122 | 0.2588 | 0.2855 |
| ✓ | ✓ | | | | 13.89 | 8.92 | 6.35 | 0.3168 | 0.2575 | 0.2871 |
| ✓ | ✓ | ✓ | | | 13.76 | 8.71 | 6.17 | 0.3146 | 0.2549 | 0.2847 |
| ✓ | ✓ | ✓ | ✓ | | **12.41** | **7.90** | 5.69 | 0.3109 | **0.2521** | 0.2815 |
| ✓ | ✓ | ✓ | ✓ | ✓ | 13.03 | 8.29 | **5.35** | **0.3090** | 0.2521 | **0.2805** |

## 4.2 ABLATION STUDY

**Effects of Proposed Components.** As shown in Tab. 4, we apply CVT without FPE and AAS to replace 3DCNN at stage-1. We observe that the depth errors are increased. Enhanced with FPE, CVT outperforms 3DCNN in the depth estimation with a large margin which showcases the significance of 3D-PE. Besides, our model benefits from the AAS, allowing our model to generalize for high-resolution images and consequently produce more accurate depth maps. Furthermore, we attribute the performance enhancements to the SVA module, which captures long-range global context information across different views to strengthen the DINOv2. Note that Norm&ALS could slightly improve the accuracy with more precise prediction. We further verify the effectiveness of these components under different image sizes in Tab. 11 of Appendix A.3 and upon other baselines in Tab. 13 of Appendix A.5. More experiment results about detailed transformer settings, qualitative visualizations, and the selection of DINOv2 layers are discussed in Appendix A.1.

**Different Attention Mechanisms for Cost Volume Regularization.** As in Tab. 5(a), linear attention suffers from terrible performance, this surprising outcome can be attributed to the nature of features within the cost volume, primarily relying on group-wise feature dot product and variance, which lacks informative representations. In contrast, even though window-based attention permits shifted-window interactions (Liu et al., 2021), it struggles to outperform vanilla attention. This indicates the crucial importance of capturing global contextual information in cost volume regularization. Besides, vanilla attention enhanced by AAS mitigates attention dilution for large image scales, leading to more accurate depth estimations and superior robustness.

**Different Attention Mechanisms for Feature Encoder.** We conduct several experiments with different types of attention during the feature extraction in Tab. 5(b). Different from the results of Tab. 5(a), linear attention outperforms other attention mechanisms in the overall error of point cloud, while the advantage of depth-related metrics is not prominent compared with vanilla attention, except for the large depth error $e_8$. We should clarify that linear attention is naturally robust for high-resolution images without attention dilution, thus it can be seen as a more reasonable and efficient choice to be applied for cross-view feature learning in SVA. For the Top-k (Zhu et al., 2023) and shifted window-based attention, they failed to achieve proper results because of a lack of global receptive fields during the feature extraction.

Table 5: We evaluate the performance of different attention mechanisms in feature encoder and cost volume regularization, including vanilla attention (with/without AAS), linear attention (Katharopoulos et al., 2020), top-k attention (Zhu et al., 2023), and shifted window attention (Liu et al., 2021).

| Cost volume attention | $e_2 \downarrow$ | $e_4 \downarrow$ | $e_8 \downarrow$ | Overall$\downarrow$ |
|-----------------------|------|------|------|---------|
| Shifted Window | 15.03 | 9.93 | 6.90 | 0.2862 |
| Linear | 15.94 | 10.64 | 7.80 | 0.2980 |
| Vanilla | 13.89 | 8.91 | 6.34 | 0.2871 |
| Vanilla + AAS | **13.76** | **8.71** | **6.17** | **0.2847** |

| Feature encoder attention | $e_2 \downarrow$ | $e_4 \downarrow$ | $e_8 \downarrow$ | Overall$\downarrow$ |
|---------------------------|------|------|------|---------|
| Shifted Window | 12.80 | 8.05 | 5.64 | 0.2862 |
| Top-K | 13.04 | 8.43 | 6.12 | 0.2854 |
| Vanilla | 12.65 | **7.88** | 5.60 | 0.2835 |
| Vanilla + AAS | **12.63** | **7.88** | 5.60 | 0.2824 |
| Linear | 13.03 | 8.29 | **5.35** | **0.2805** |

## 5 CONCLUSION

In this paper, we delve into the attention mechanisms within the feature encoder and cost volume regularization of the MVS pipeline. Our model seamlessly incorporates cross-view information to pre-trained DINOv2 features through SVA. Moreover, we propose specially designed FPE and AAS to strengthen the ability of CVT to generalize high-resolution images. The proposed MVSFormer++ can achieve state-of-the-art results in DTU and rank top-1 on Tanks-and-Temples.

## ACKNOWLEDGMENTS

This work was supported by the Shanghai Municipal Science and Technology Major Project (2021SHZDZX0103), Shanghai Platform for Neuromorphic and AI Chip under Grant 17DZ2260900 (NeuHelium), and the computations in this work were performed using the CFFF platform of Fudan University.

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

# A APPENDIX

## A.1 MORE DETAILS AND EXPERIMENTS OF TRANSFORMERS IN MVS

**Sequential Length of CVT.** The sequential lengths of CVT's feature are downsampled to $[H_0/32, W_0/32, D = 16]$, where $H_0, W_0$ mean the original image height and width respectively.

DTU images are tested in 1152×1536, resulting in the sequential length of 36*48*16 = 27,648 to CVT. Tanks-and-Temple images are tested in 1088×1920, resulting in the sequential length of 34*60*16 = 32,640. Our multi-scale training is based on Cao et al. (2022), *i.e.*, from 512 to 1280, resulting in sequential length from 5,120 to 20,480. We set the average length $\overline{n}$ =12,185.

**Multi-layer Features of DINOv2 and SVA.** Fig. 6(a) shows the logarithmic absolute mean and maximum values of DINOv2 features from different layers, while Fig. 6(b) illustrates the zero-shot depth estimation based on Winner-Take-All (WTA) feature correlation (Collins, 1996) of different layers' DINOv2 features[2]. Generally, middle layers enjoy better zero-shot performance. We also provide ablation studies about the usage of different feature layers in Tab. 6. To efficiently explore the layer-selecting strategy, we empirically adopt the last layer (11) of DINOv2. Because dense vision tasks in Oquab et al. (2023) ensure the robust performance of the last layer of DINOv2. Subsequently, we abandon shallow layers in DINOv2 (0,1) which contains more positional information rather than meaningful features (Amir et al., 2021; Walmer et al., 2023). Thus we uniformly sample among mid-layer and last-layer features from DINOv2 in Tab. 6, and find a relatively good combination, *i.e.*, (3,7,11). Though the 5th layer shows better zero-shot WTA feature correlations in Fig. 6(b), the gap between (3,5,11) and (3,7,11) is not obvious. As shown in Fig. 6(a), layers from 8 to 10 are unstable, which would cause inferior results to the 4-layer setting.

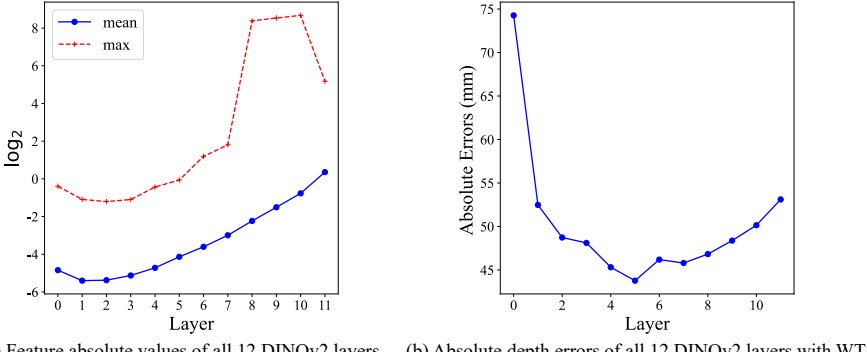

(a) Feature absolute values of all 12 DINOv2 layers.     (b) Absolute depth errors of all 12 DINOv2 layers with WTA.

Figure 6: Detailed analysis of DINOv2 features: (a) absolute mean and maximum values of DINOv2 features; (b) absolute depth errors achieved from different DINOv2 layers with WTA (Collins, 1996).

**Pre-LN vs Post-LN.** As shown in Tab. 6, for the feature encoder, we compare the placement of LN between Pre-LN and Post-LN (Wang et al., 2019), which can be formulated as:

$$\begin{aligned} \text{Pre-LN}(x) : & x = x + \text{Attn}(\text{LN}(x)), x = x + \text{FFN}(\text{LN}(x)), \\ \text{Post-LN}(x) : & x = \text{LN}(x + \text{Attn}(x)), x = \text{LN}(x + \text{FFN}(x)), \end{aligned} \tag{3}$$

where $\text{LN}(\cdot)$, $\text{Attn}(\cdot)$, $\text{FFN}(\cdot)$ indicate layer normalization, Attention and FFN blocks respectively. We find that Pre-LN achieves smaller depth errors than Post-LN, which is different from many NLP tasks that Post-LN reaches better performance than Pre-LN. This is due to the large variance of DINOv2 features and faster convergence of Pre-LN. While in cost volume regularization, Post-LN outperforms the Pre-LN.

**Validation Logs with/without Norm&ALS.** We compare the validation logs during the training phase of MVSFormer++ with/without Pre-LN and ALS in Fig. 7. Two models are trained for 15 epochs at all. The model with the combination of Pre-LN and ALS enjoys better convergence and achieves the best checkpoint in epoch 10, which is 4 epochs earlier than the one with Post-LN and no ALS. Note that the former also enjoys better performance in the point cloud reconstruction.

**Different Variants of MLP in Transformer.** Although the transformer with GLU (Shazeer, 2020) leads to smaller depth errors, it still struggles to outperform the original FFN in Tab. 6. For a more general architecture, we still use FFN in our final design.

---

[2]WTA used in DINOv2 means directly calculating feature correlations from a certain layer of DINOv2 along the epipolar line and deciding the depth estimation according to the highest feature similarity.

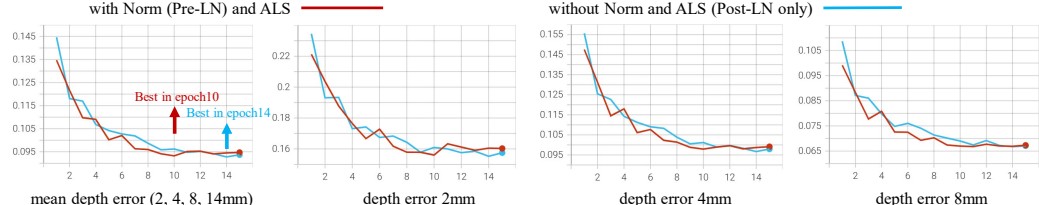

Figure 7: Validation logs with/without Norm&ALS during the training phase. MVSFormer++ achieves the best checkpoint in epoch 10 with Pre-LN and ALS, which is 4 epochs earlier than the one with Post-LN and no ALS.

**Qualitative Ablation Study.** We show qualitative ablation studies in Fig. 8. From the visualization, CVT and AAS could effectively eliminate the outliers (Fig. 8(c)). SVA with normalized 2D-PE is also critical for precise point clouds (Fig. 8(e)).

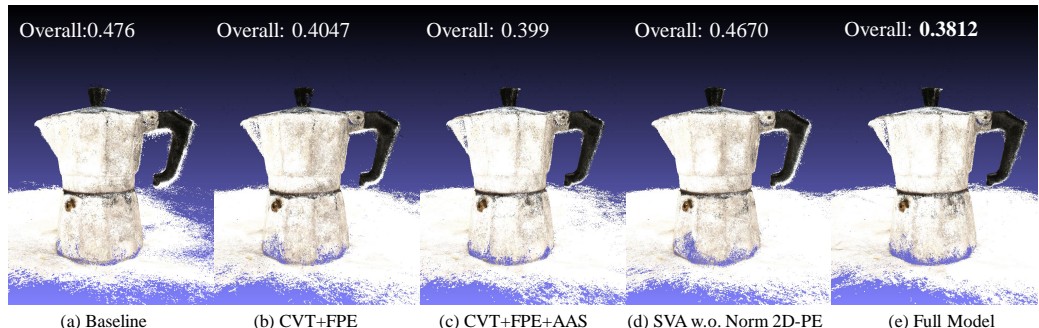

Figure 8: Qualitative comparisons for the ablation studies of MVSFormer++.

**Ablations about Normalized 2D-PE.** We use normalized 2D-PE to replace the vanilla PE in the baseline model (Ding et al., 2022). Tab. 7 shows that the baseline with normalized 2D-PE outperforms the one with standard 2D-PE.

**The Selection of DINOv2.** Our choice of DINOv2 (Oquab et al., 2023) as the backbone of MVS-Former++ is attributed to its robust zero-shot feature-matching capability and impressive performance in dense vision downstream tasks. Compared to other ViTs, DINOv2 was trained on large-scale curated data, unifying both image-level (contrastive learning (Caron et al., 2021)) and patch-level (masked image prediction (He et al., 2022)) objectives, as well as high-resolution adaption. Hence DINOv2 enjoys remarkably stable feature correlation across different domains. Moreover, DINOv2 achieves better performance in dense vision benchmarks, such as semantic image segmentation and monocular depth estimation. Our experiments in Tab. 8 show the efficacy of DINOv2 in MVS. Note that the frozen DINOv2 even achieves slightly better performance compared to the trainable Twins (Chu et al., 2021). We further explore the low-rank fine-tuning version (LoRA (Hu et al., 2021), rank 16) of DINOv2, but the improvements are not very significant with expanded computation. Besides, the full-model fine-tuned DINOv2 performs worse than the frozen one, which also makes sense. Since DINOv2 is a robust model pre-trained on large-scale curated data, the MVS fine-tuning is based on the limited DTU, which degrades the generalization of DINOv2's features. Thus, we use frozen DINOv2-base as a strong and robust baseline of MVSFormer++ in Tab. 4.

## A.2 INFERENCE MEMORY AND TIME COSTS

We evaluate the memory and time costs for inference when using images with a resolution of $1152 \times 1536$ as input. All comparisons are based on NVIDIA RTX A6000 GPU, From Tab. 9, MVSFormer++ is slightly faster than MVSFormer (Cao et al., 2022), owing to the efficiency of FlashAttention during the inference. For other learning-based methods, TransMVSNet (Ding et al., 2022) suffers from long inference time. GeoMVSNet (Zhang et al., 2023c) have large parameters due to the heavy architecture of heavy cost volume regularization and feature fusion module. We also evaluate the memory cost compared to CasMVSNet (Gu et al., 2020) with different image resolutions and depth intervals in Tab. 10. Besides the FlashAttention (Dao, 2023), MVSFormer++ enjoys a more

Table 6: We evaluate some detailed designs including the number of cross-view layers in SVA, the usage of different DINOv2's feature layers (0∼11), the placement of LN in the attention mechanism, and different variants of MLP after the attention in feature encoder and cost volume regularization.

| Feature Encoder | | | | Cost Volume | | Metric | | | |
|---|---|---|---|---|---|---|---|---|---|
| Layer nums | DINOv2 layers | Norm | MLP | Norm | MLP | $e_2 \downarrow$ | $e_4 \downarrow$ | $e_8 \downarrow$ | Overall$\downarrow$ |
| 2 | (5,11) | Pre-LN | FFN | Post-LN | FFN | 13.84 | 8.99 | **5.14** | 0.2836 |
| 3 | (3,5,11) | Pre-LN | FFN | Post-LN | FFN | 12.94 | 8.08 | 5.59 | 0.2806 |
| 3 | (3,7,11) | Pre-LN | FFN | Post-LN | FFN | 13.03 | 8.29 | 5.35 | **0.2805** |
| 4 | (2,5,8,11) | Pre-LN | FFN | Post-LN | FFN | 13.20 | 8.50 | 6.97 | 0.2832 |
| 3 | (3,7,11) | Pre-LN | FFN | Pre-LN | FFN | 13.46 | 8.74 | 6.14 | 0.2827 |
| 3 | (3,7,11) | Post-LN | FFN | Post-LN | FFN | 13.30 | 8.99 | **5.14** | 0.2850 |
| 3 | (3,7,11) | Pre-LN | GLU | Post-LN | GLU | **11.95** | **7.81** | 5.79 | 0.2809 |

Table 7: Ablation results of normalized 2D PE in FMT module of TransMVSNet.

| Normalized 2D-PE | $e_2 \downarrow$ | $e_4 \downarrow$ | $e_8 \downarrow$ |
|---|---|---|---|
| × | 23.92 | 19.54 | 16.36 |
| ✓ | **23.15** | **18.68** | **15.35** |

Table 8: Quantitative comparisons of MVSFormer (Cao et al., 2022) based on different ViT backbones on DTU dataset. Results of DINO and Twins are from Cao et al. (2022).

| ViT backbone | Frozen backbone | Accuracy$\downarrow$ | Completeness$\downarrow$ | Overall$\downarrow$ |
|---|---|---|---|---|
| DINO-small | ✓ | 0.327 | 0.265 | 0.296 |
| DINO-base | ✓ | 0.334 | 0.268 | 0.301 |
| Twins-small | × | 0.327 | 0.251 | 0.289 |
| Twins-base | × | 0.326 | 0.252 | 0.289 |
| DINOv2-base | ✓ | **0.3198** | 0.2549 | 0.2875 |
| DINOv2-base | LoRA (rank=16) | 0.3239 | **0.2501** | **0.2870** |
| DINOv2-base | × | 0.3244 | 0.2566 | 0.2905 |

reasonable depth hypothesis setting (32-16-8-4 vs 48-16-8) compared with CasMVSNet, which leads to lower memory cost. Hence MVSFormer++ contains sufficient scalability for dense and precise depth estimations.

## A.3 Ablation Study about Different Image Sizes

To confirm the robustness of transformer blocks in MVSFormer++ across different image sizes, we further expand the ablation studies in Tab. 4 to Tab. 11. These ablations demonstrate the effectiveness of FPE, AAS, and normalized 2D-PE. Generally, both high-resolution depth and point cloud results outperform the low-resolution ones. This conclusion is the same as most MVS methods (Giang et al., 2022; Cao et al., 2022), which proves the importance of adaptability for high-resolution images. Moreover, FPE and AAS improve the results under both $576 \times 768$ and $1152 \times 1536$ images. However, AAS is more effective for depth estimation in high-resolution cases, while the depth gap in low-resolution ones is not pronounced. Most importantly, the normalized 2D-PE plays a very important role in high-resolution feature encoding, contributing substantial improvements in both depth and point clouds.

## A.4 Quantitative results on ETH3D datasets.

To demonstrate MVSFormer++'s good generalization on large-scale scenes with high-resolution images, we evaluate MVSFormer++ on the test set of ETH3D which is trained on a mixed DTU and BlendedMVS dataset. We resize images to 2k, following the setting of Tanks-and-Temple, while 10 views are involved. Quantitative comparisons on ETH3D are shown in Tab. 12, compared with other state-of-the-art learning-based methods and traditional ones, our method achieves the best performance in the ETH3D dataset. Note that our ETH3D results are all achieved with the

Table 9: Illustration of model memory and time costs during the inference phase of $1152 \times 1536$ images and model parameters (Params.).

| | Memory (MB) | Time (s/img) | Params. (all) | Params. (trainable) |
|---|---|---|---|---|
| MVSFormer (Cao et al., 2022) | 4970 | 0.373 | 28.01M | 28.01M |
| MVSFormer++ (Ours) | 5964 | 0.354 | 126.95M | 39.48M |
| CasMVSNet (Gu et al., 2020) | 6672 | 0.4747 | 0.93M | 0.93M |
| TransMVSnet (Ding et al., 2022) | 6320 | 1.49 | 1.15M | 1.15M |
| GeoMVSNet (Zhang et al., 2023c) | 9189 | 0.369 | 15.31M | 15.31M |

Table 10: Comparison of GPU memory between MVSFormer++ and CasMVSNet with different resolutions and depth intervals.

| Methods | Resolution | Depth Interval | Memory (MB) |
|---|---|---|---|
| CasMVSNet | $864 \times 1152$ | | 4769 |
| | $1152 \times 1536$ | 48-32-8 | 6672 |
| | $1088 \times 1920$ | | 7659 |
| MVSFormer++ | $864 \times 1152$ | 32-16-8-4 | 4873 |
| | | 64-32-8-4 | 5025 |
| | $1152 \times 1536$ | 32-16-8-4 | 5964 |
| | | 64-32-8-4 | 6753 |
| | $1088 \times 1920$ | 32-16-8-4 | 6613 |
| | | 64-32-8-4 | 7373 |

Table 11: Ablation studies of MVSFormer++ under different image scales on DTU.

| Resolution | FPE | AAS | SVA 2D-PE | Norm 2D-PE | $e_2 \downarrow$ | $e_4 \downarrow$ | $e_8 \downarrow$ | Accuracy$\downarrow$ | Completeness$\downarrow$ | Overall$\downarrow$ |
|---|---|---|---|---|---|---|---|---|---|---|
| $576 \times 768$ | | | | | 20.85 | 14.03 | 9.28 | 0.3783 | 0.3060 | 0.3422 |
| | ✓ | | | | 17.99 | 11.06 | 7.49 | 0.3649 | 0.3116 | 0.3383 |
| | ✓ | ✓ | | | 17.97 | 11.13 | 7.54 | 0.3571 | 0.3126 | 0.3348 |
| | ✓ | ✓ | ✓ | | 17.28 | 11.01 | 7.56 | 0.3512 | 0.3112 | 0.3312 |
| | ✓ | ✓ | | ✓ | 17.21 | 10.94 | 7.47 | 0.3504 | 0.3098 | 0.3301 |
| $1152 \times 1536$ | | | | | 16.38 | 11.16 | 7.79 | 0.3198 | 0.2549 | 0.2875 |
| | ✓ | | | | 13.89 | 8.92 | 6.35 | 0.3168 | 0.2575 | 0.2871 |
| | ✓ | ✓ | | | 13.76 | 8.71 | 6.17 | 0.3146 | 0.2549 | 0.2847 |
| | ✓ | ✓ | ✓ | | 15.00 | 9.74 | 6.50 | 0.3330 | **0.2514** | 0.2922 |
| | ✓ | ✓ | | ✓ | **13.03** | **8.29** | **5.35** | **0.3090** | 0.2521 | **0.2805** |

same threshold (0.5) of depth confidence filter and default settings of dynamic point cloud fusion (DPCD) (Yan et al., 2020), without any cherry-pick hyper-parameter adjusting. This robust and impressive performance can be attributed to the effectiveness of our meticulously designed positional components for transformers, such as normalized 2D-PE, FPE, and AAS. These techniques enable our method to effectively generalize to various image scales.

Table 12: Quantitative results on ETH3D datasets.

| Methods | Precision↑ | Recall↑ | F1-Score↑ |
|---|---|---|---|
| Gipuma (Galliani et al., 2015a) | 86.47 | 24.91 | 45.18 |
| PMVS (Furukawa & Ponce, 2009) | 90.08 | 31.84 | 44.16 |
| COLMAP (Schönberger et al., 2016) | **91.97** | 62.98 | 73.01 |
| CostFormer (Chen et al., 2023) | - | - | 77.36 |
| MVSFormer (Cao et al., 2022) | 82.23 | 83.75 | 82.85 |
| MVSFormer++ (Ours) | 81.88 | **83.88** | **82.99** |

## A.5 EFFECT OF PROPOSED COMPONENTS BASED ON OTHER BASELINES

We present quantitative ablation studies in Tab. 13 for other baselines enhanced by our components. These previous MVS methods include CasMVSNet (Gu et al., 2020) and MVSFormer + DINOv1 (Caron et al., 2021) (MVSFormer-P (Cao et al., 2022)). Specifically, we re-train CasMVS-Net (CasMVSNet*) as an intermediate baseline for a fair comparison, which contains a 4-stage depth hypothesis setting (32-16-8-4) and cross-entropy loss, sharing the same setting with MVSFormer and MVSFormer++. Since the proposed SVA is a side-tuning module specifically designed for pre-trained models, we only evaluate the effect of SVA on MVSFormer-P. From Tab. 13, our CVT demonstrates substantial improvements for both CasMVSNet* and MVSFormer-P, and our SVA further enhances the results of MVSFormer-P with CVT.

Table 13: Quantitative ablation studies of CasMVSNet (Gu et al., 2020) and MVSFormer + DI-NOv1 (Caron et al., 2021) (MVSFormer-P) based on our proposed components including CVT and SVA. * indicates that CasMVSNet is re-trained with the 4-stage depth hypothesis setting (32-16-8-4) and cross-entropy loss as MVSFormer (Cao et al., 2022) and MVSFormer++.

| Methods | $e_2 \downarrow$ | $e_4 \downarrow$ | $e_8 \downarrow$ | Accuracy$\downarrow$ | Completeness$\downarrow$ | Overall$\downarrow$ |
|---|---|---|---|---|---|---|
| CasMVSNet | 30.21 | 24.63 | 21.14 | 0.325 | 0.385 | 0.355 |
| CasMVSNet* | 23.15 | 18.68 | 15.35 | 0.353 | 0.286 | 0.320 |
| CasMVSNet* + CVT | 15.70 | 10.13 | 7.14 | 0.332 | 0.278 | 0.305 |
| MVSFormer-P | 17.18 | 11.96 | 8.53 | 0.327 | 0.265 | 0.296 |
| MVSFormer-P + CVT | 14.25 | 9.13 | 6.51 | 0.327 | **0.261** | 0.294 |
| MVSFormer-P + CVT + SVA | **13.55** | **8.67** | **6.31** | **0.322** | 0.254 | **0.288** |

## A.6 MORE RESULTS ON DTU DATASET

We show the qualitative depth comparisons between MVSFormer and MVSFormer++ in Fig. 9. Our method estimates more precise depth maps even in challenging scenes. Fig. 12 shows all predicted point clouds on the DTU test set. These point clouds are accurate and complete, especially in the textureless regions.

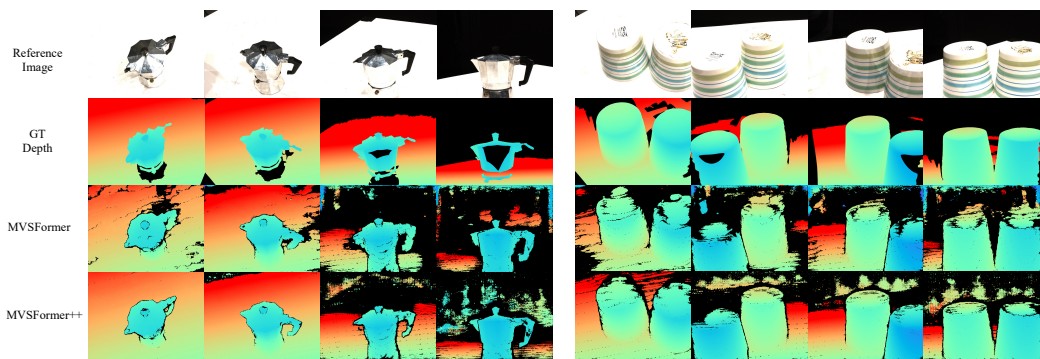

Figure 9: Qualitative depth comparisons on DTU between MVSFormer (Cao et al., 2022) and MVSFormer++.

## A.7 MORE RESULTS ON TANKS-AND-TEMPLES

We show some qualitative comparisons between MVSFormer and MVSFormer++ in Fig. 10. MVS-Former++ achieves geometric reconstructions with better precision, while MVSFormer shows more complete results in some scenes, such as the "Recall" of "Playground". For the trading-off between precision and recall, MVSFormer++ obviously enjoys a superior balance. Point cloud results of the Intermediate and Advanced set are shown in Fig. 13.

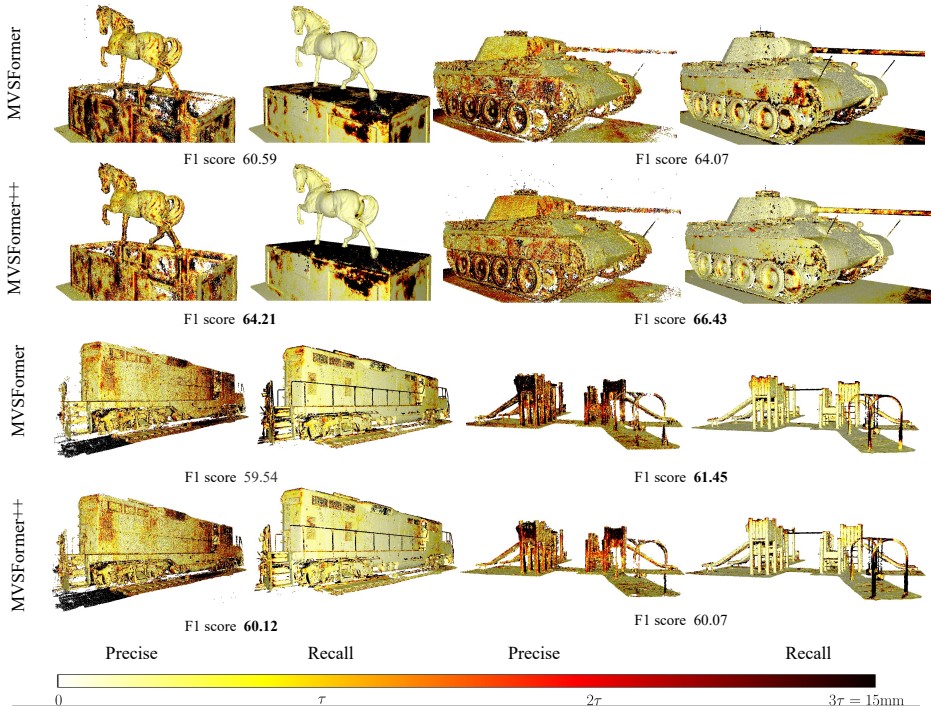

Figure 10: Qualitative results of Tanks-and-Temples (Horse, Panther, Train, and Playground) compared between MVSFormer and MVSFormer++. $\tau$ is the threshold to measure errors which are set officially to 3mm, 5mm, 5mm, and 10mm for these scenes respectively.

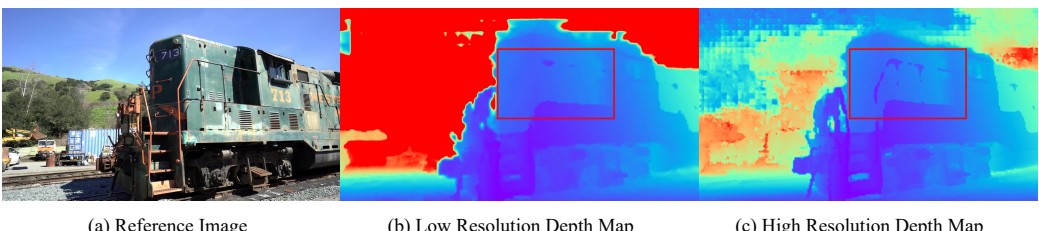

(a) Reference Image      (b) Low Resolution Depth Map      (c) High Resolution Depth Map

Figure 11: Depth map of "Train" in Tanks-and-Temples with half and full resolution images as inputs. For the low-resolution depth, some railings are missed because of the error accumulation of multi-stage architecture.

## A.8 LIMITATION AND FUTURE WORKS.

Though MVSFormer++ enjoys powerful MVS capability as well verified in our experiments, it still suffers from similar limitations as other coarse-to-fine MVS models. Specifically, the coarse stage struggles for inevitable error estimations for tiny foregrounds, resulting in error accumulations for the following stages as shown in Fig. 11. Designing a novel dynamic depth interval selection strategy would be a potential solution to handle this problem. Since some work (Yang et al., 2022) have investigated this issue, combining them with our work could be seen as interesting future work.

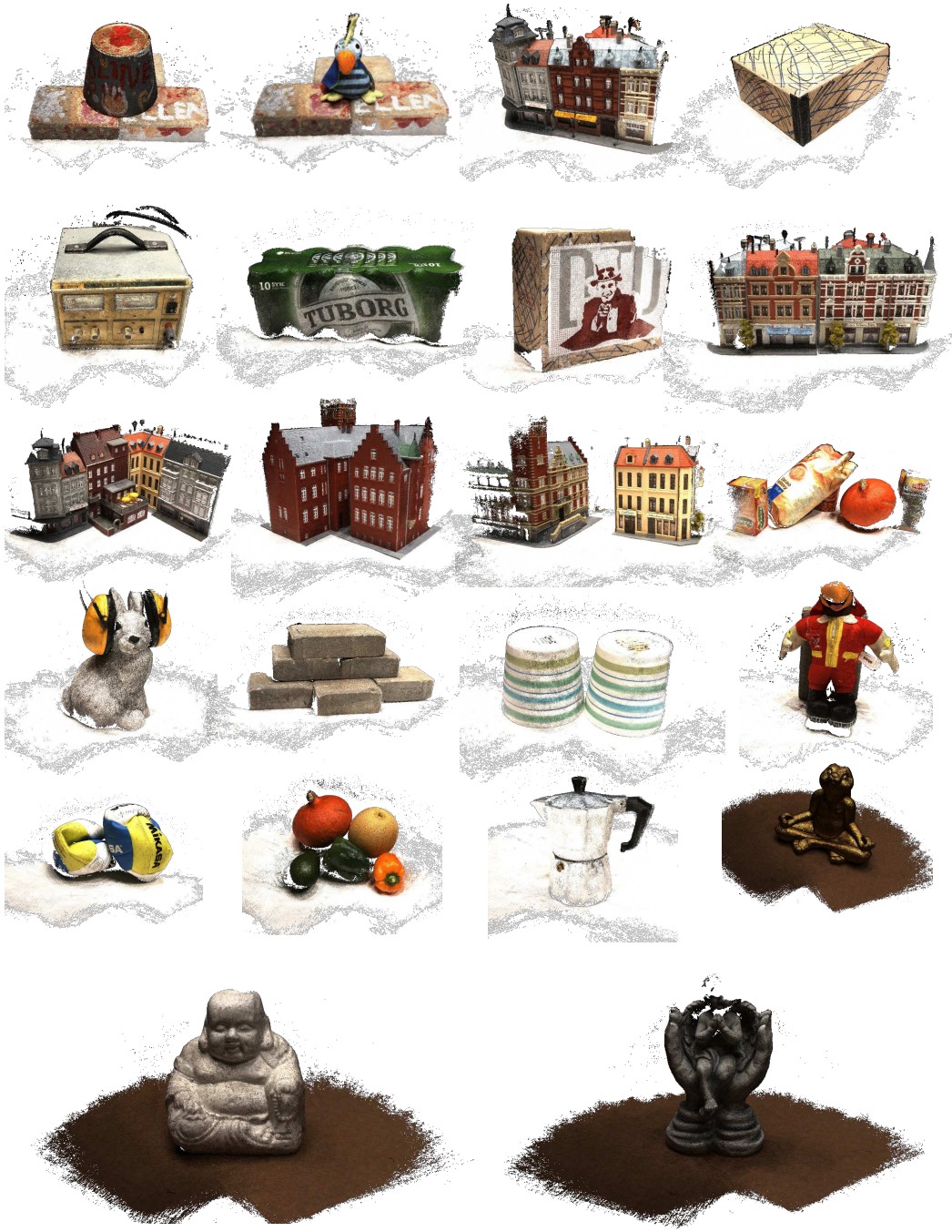

Figure 12: Point cloud results of DTU (Aanæs et al., 2016).

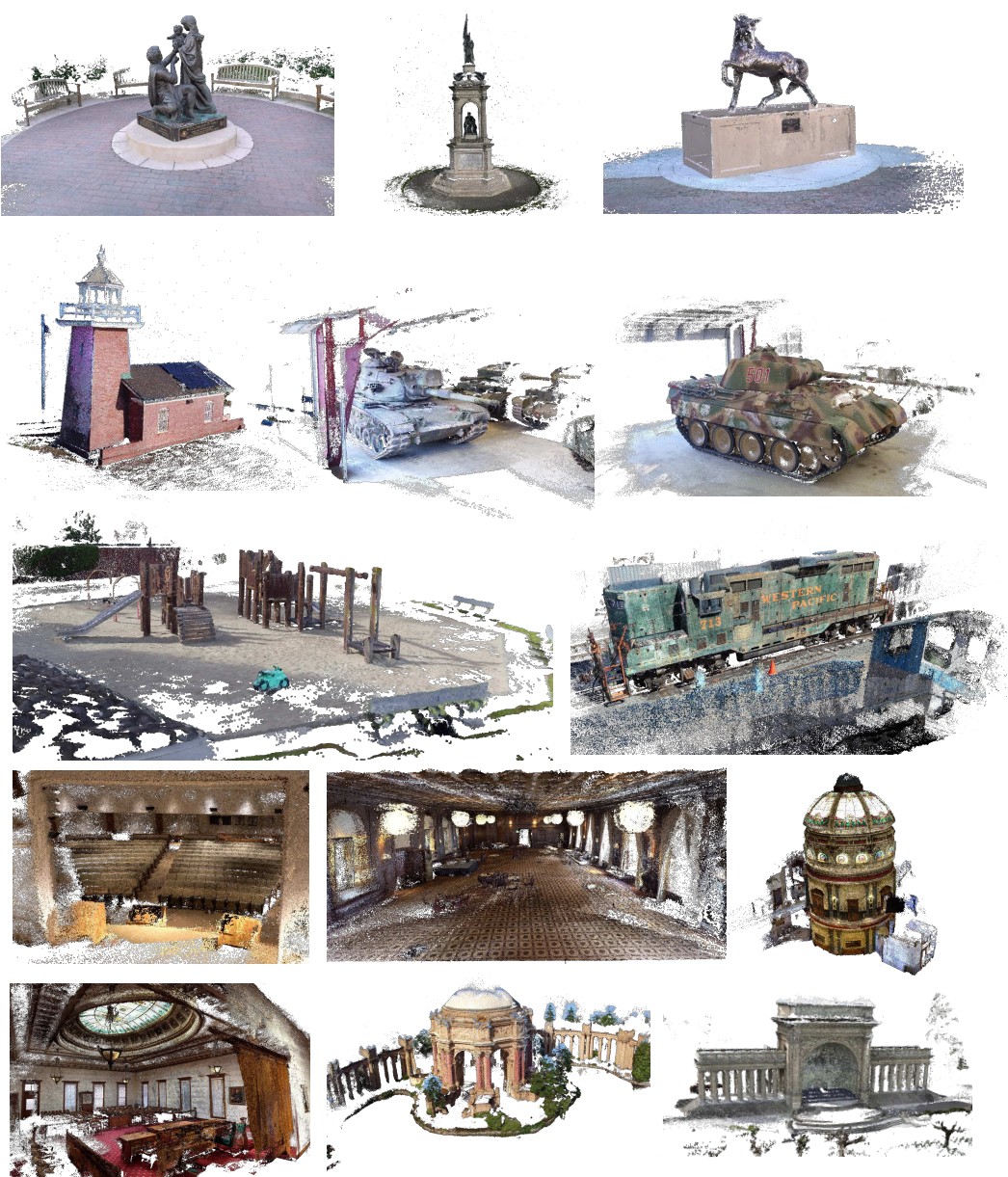

Figure 13: Point cloud results of Tanks-and-Temples (Knapitsch et al., 2017).

