# OpenReview forum: "MVSFormer++: Revealing the Devil in Transformer's Details for Multi-View Stereo"
_ICLR.cc/2024/Conference — ICLR 2024 poster_

### Official Review · Reviewer_Xp5d · 2023-10-29

**Soundness:** 3 good
**Presentation:** 3 good
**Contribution:** 2 fair
**Rating:** 6
**Confidence:** 5

**Summary:**

The paper proposes MVSFormer++, which is an extended/enhanced version of the previous work MVSFormer. The authors have well-studied the usage of the transformer at different stages of the learning-based MVS pipeline, and have demonstrated the effectiveness of the proposed components by extensive experiments. The proposed pipeline achieves SOTA results on several MVS datasets.

**Strengths:**

- The method achieves SOTA results on DTU, Tanks and Temples, and ETH3D datasets. I believe currently it is one of the best-performing MVS approaches.

- The authors have conducted extensive experiments to demonstrate the effectiveness of the proposed components. I can find ablation studies on each proposed component in the experimental section.

**Weaknesses:**

- The paper proposed a bunch of small components/tricks over the previous MVSFormer. I acknowledge that these tricks might be useful, however, I would feel like each of them is a bit incremental and the whole story is not that interconnected. For the conference paper, I prefer a neat idea/key component that can bring strong improvements. The paper looks more like an extended journal version paper of the previous one.

- ETH3D evaluation: the proposed method does not perform well on ETH3D even compared with other learning-based approach (e.g., Vis-MVSNet, EPP-MVSNet). Could the authors explain potential causes?

- These is no a limitation section. I would like know the scalability of the proposed method, will the memory cost dramatically increased compared with CNN based approaches (e.g., CasMVSNet) when the image size/depth sample number increase?

**Questions:**

See above.

---

> ### Author Response · Authors · 2023-11-16
> **Response to Reviewer Xp5d**
>
> Thanks for the valuable feedback on our work. We have thoroughly investigated that MVSFormer++ enjoys good scalability. And we would like to engage in further discussions to address any confusion.
>
> **1.Incremental ideas rather than neat ones with strong improvements?**
>
> Thanks for this comment. We would like to clarify that our contributions are not merely incremental. The decision to build upon MVSFormer is strategic, as it serves as a simple yet highly effective baseline with strong scalability and significant potential. With the development of pre-trained ViTs and the widespread adoption of transformers in feature-matching tasks, as highlighted in the second paragraph of the introduction, exploring the reasonable usage of transformers in MVS becomes very important. MVSFormer++ systematically addressed three key challenges that hinder effective transformer-based learning in MVS (already previously outlined in the third paragraph of the introduction):
>
> 1) *Tailored attention mechanisms for different MVS modules.* We provide insight indicating that linear attention is well-suited for feature encoding with superior robustness in terms of sequential length, while cost volume regularization benefits more from spatial relations, favoring vanilla and window-based attention mechanisms.
>
> 2) *Incorporating cross-view information into Pre-trained ViTs.* Our Side View Attention (SVA) efficiently learns cross-view information through frozen DINOv2.
>
> 3) *Enhancing Transformer’s Length Extrapolation Capability in MVS.* We effectively address the length extrapolation issue through normalized 2D-PE, Frustoconical Positional Encoding (FPE), and Adaptive Attention Scaling (AAS) as verified in Table12.
>
> Moreover, our neat and simple combination of Cost Volume Transformer (CVT) enhanced with FPE and AAS achieves substantial improvements based on not only MVSFormer but also CasMVSNet as shown in Table13 of Appendix A.5.
> We anticipate that our work could benefit the community by bridging the gap between MVS learning and reliable transformer utilization.
>
> **2.ETH3D evaluation is not competitive enough when compared to Vis-MVSNet and EPP-MVSNet.**
>
> Thanks for this point. We need to clarify that our ETH3D results are all achieved with the same threshold (0.5) of depth confidence filter and default settings of dynamic point cloud fusion (DPCD)[1] without any cherry-pick hyper-parameter adjusting. This approach ensures a fair and unbiased evaluation. Though improvements may not be very significant, they are consistent and indicative of the robustness and competitiveness of MVSFormer++.  We have claimed these points in the paper revision. For the results of Vis-MVSNet and EPP-MVSNet, unfortunately, we could not find their detailed hyper-parameter settings for ETH3D to re-implement their results. Since our goal is to propose a generalized and powerful MVS method, we focus on the intrinsical depth estimation quality rather than sophisticated post-precessing tricks. Our codes and evaluation scripts will be fully open-released if accepted.
>
> [1] Yan J, Wei Z, Yi H, et al. Dense hybrid recurrent multi-view stereo net with dynamic consistency checking. ECCV2020.
>
> **3.Scalability: how does the memory cost increase when the image/depth sample number increases?**
>
> Thanks for this insightful comment. We have investigated the scalability in both image and depth sizes and compared MVSFormer++ to CasMVSNet in Table11. Benefiting from an efficient attention implementation[2] and a more reasonable 4-stage depth hypothesis setting, MVSFormer++ demonstrates robust scalability for dense and precise depth estimation. The development of the highly optimized attention mechanism also improves the significance of our work in advancing the effective utilization of transformers in MVS. Moreover, our CVT enhanced by FPE and AAS could also be well extended into large-scale images, which overcomes the key challenge of previous transformer-based manners.
>
> [2] Dao T. Flashattention-2: Faster attention with better parallelism and work partitioning[J]. arXiv preprint arXiv:2307.08691, 2023.
>
> **4.Missing discussion about limitation**
>
> Thanks for this point. We have provided related discussions in Appendix A.8. Though MVSFormer++ enjoys powerful MVS capability as well verified in our experiments, it still suffers from similar limitations as other coarse-to-fine MVS models.
> Specifically, the coarse stage struggles for inevitable error estimations for tiny foregrounds in low-resolution inputs, resulting in error accumulations for the following stages as shown in Figure12. Designing a novel dynamic depth interval selection strategy would be a potential solution to handle this problem, which can be seen as interesting future work.

---

### Official Review · Reviewer_4iWa · 2023-10-30

**Soundness:** 3 good
**Presentation:** 3 good
**Contribution:** 3 good
**Rating:** 5
**Confidence:** 3

**Summary:**

This paper introduces MVSFormer++, a learning-based Multi-View Stereo (MVS) method that leverages pre-trained models to enhance depth estimation in MVS. The study tackles a crucial gap in existing research by exploring the impact of transformers on various MVS modules. The paper's motivation is clear. However, there are notable areas that require attention for improvement.

**Strengths:**

The paper innovatively introduces transformer-based models and attention mechanisms to address a vital issue in MVS. The novelty lies in the thorough exploration of different transformer attention mechanisms across diverse MVS modules. The paper provides hypotheses and experimental evidence supporting the use of different attention mechanisms in the feature encoder and cost volume regularization.

The authors conducted experiments across multiple benchmarks, including DTU, Tanks-and-Temples, BlendedMVS, and ETH3D, showcasing MVSFormer++'s state-of-the-art performance on challenging benchmarks (DTU and Tanks-and-Temples). This highlights the practical significance of the proposed approach.

The paper includes well-executed ablation studies, comparing the impacts of different attention mechanisms on various MVS modules.

**Weaknesses:**

(1) Clarity and Detail:
The paper lacks detailed explanations of specific design choices, such as the rationale behind selecting DINOv2 as the base model. Additionally, the utilization of different levels of DINOv2 features is not clearly elucidated. It is recommended to include these details to enhance the manuscript's clarity and independence.

(2) Experiments:
While the incorporation of DINOv2 in the feature extraction stage significantly enhances performance, it is crucial to clarify that this improvement is not solely due to the increase in the number of parameters. The improvement in point cloud evaluation metrics by the proposed module during ablation experiments appears subtle. To bolster the paper's experimental support, I recommend validating the proposed module's effectiveness by integrating it into baseline methods and conducting a comparative analysis. This would provide a clearer understanding of the module's actual contribution.

Additionally, it is advisable to explore more pre-trained models and conduct ablation experiments without pre-trained models. Given DINOv2's frozen state during training, fine-tuning it serves as a pivotal baseline.

Furthermore, the paper should include visual comparisons of depth maps to visually demonstrate the accuracy advantages of the estimated depth maps.

(3) Discussion of Limitations:
The paper lacks a discussion of the limitations and failure cases of MVSFormer++. Understanding the method's limitations is crucial for evaluating its real-world applicability.

**Questions:**

(1) DINOv2 Pre-training Choice:
What motivated the decision to freeze DINOv2 during pre-training? How does it uniquely contribute to your method? Would including experiments with different pre-trained models or fine-tuning DINOv2 serve as valuable comparisons?

(2) Cost-Benefit Analysis of DINOv2:
Considering the marginal improvement in point cloud metrics, is the increase in network parameters due to adopting DINOv2 justified? How can you demonstrate that the metric enhancements stem from the introduced module's contribution rather than a mere increase in parameters?

(3) Discussion of MVSFormer++ Limitations:
Could you briefly discuss MVSFormer++'s limitations, especially in scenarios where it might underperform?

---

> ### Author Response · Authors · 2023-11-16
> **Response to Reviewer 4iWa (part1/2)**
>
> Thanks for the valuable feedback on our work. We would like to clarify that our proposed components enjoy substantial improvements upon other baselines. And we would like to engage in further discussions to address any confusion.
>
> **1.More elucidation about the selection of DINOv2 and the utilization of different levels of DINOv2.**
>
> Thanks for your insightful comments.  We apologize for the insufficient discussion about DINOv2. We have provided more discussions about the usage of DINOv2 and its feature layers in Sec3 Preliminary and Appendix A.1 in our paper revision.
> Our choice of DINOv2[1] as the backbone of MVSFormer++ is attributed to its robust zero-shot feature-matching capability and impressive performance in dense vision downstream tasks. Compared to other ViTs, DINOv2 was trained on large-scale curated data, unifying both image-level (contrastive learning) and patch-level (masked image prediction) objectives, as well as high-resolution adaption. Hence DINOv2 enjoys remarkably stable feature correlation across different domains and better performance in dense vision benchmarks (segmentation and monocular depth estimation). Our experiments in Table10 (listed below) show the efficacy of DINOv2 in MVS.
>
> | ViT backbone | Frozen backbone | Accuracy$\downarrow$ | Completeness$\downarrow$ | Overall$\downarrow$ |
> |--------------|-----------------|----------------------|--------------------------|------------------------------------|
> | DINO-small   | $\checkmark$    | 0.327                | 0.265                    | 0.296               |
> | DINO-base    | $\checkmark$    | 0.334                | 0.268                    | 0.301               |
> | Twins-small  | $\times$        | 0.327                | 0.251                    | 0.289               |
> | Twins-base   | $\times$        | 0.326                | 0.252                    | 0.289               |
> | DINOv2-base  | $\checkmark$    | **0.3198**     | 0.2549                   | 0.2875              |
> | DINOv2-base  | LoRA (rank=16)  | 0.3239               | **0.2501**          | **0.2870**     |
>
> For the selection of DINOv2 layers, we have carefully analyzed the feature characteristics in Figure6 and Table6 of the Appendix. Here we provide more detailed discussions as follows. We show the WTA depth estimations of DINOv2 in Figure6(b), which indicate middle layers are more useful to MVS. So we empirically compare the layer ablations in Table6 to verify our claims. Generally, our final combination (3,7,11) is a good selection. Though the 5th layer shows better zero-shot WTA feature correlations in Figure6(b), the gap between (3,5,11) and (3,7,11) is not obvious. As shown in Figure6(a), layers from 8 to 10 are unstable, which would cause inferior results to the 4-layer setting. More details are discussed in Appendix A.1.
>
> We should clarify that the usage of DINOv2 and other ViT backbones are not the main contributions of this paper. Our improvements are not solely dependent on the increased capacity of pre-trained ViTs. We will explain this in the next question.
>
> [1] Oquab M, Darcet T, Moutakanni T, et al. Dinov2: Learning robust visual features without supervision[J]. arXiv preprint arXiv:2304.07193, 2023.
>
> **2.The improvement is subtle when compared to the usage of DINOv2.**
>
> Thanks for this point.  Although MVSFormer+DINOv2 is a strong baseline (Table10), we need to clarify that our work is orthogonal to the DINOv2. DINOv2 just improved the overall distance from 0.289 to 0.2875 with less computation (frozen backbone compared with Twins), while we further improved the overall distance from 0.2875 to 0.2805 with our novel designs of SVA, CVT, and image scaling adaption.
>
> **3.DINOv2 fine-tuning result as the pivotal baseline.**
>
> Thanks. As discussed in MVSFormer[2], fine-tuning a Plain ViT (DINO, DINOv2) is very costly, especially for high-resolution images. Thus we try to optimize the low-rank fine-tuning (LoRA[3]) rank=16 as a new baseline to verify whether more trainable parameters of the ViT backbone result in prominent improvements (Table10).
> The result indicates that more trainable parameters in the backbone only achieve minor improvements. This result also proves the effectiveness of our proposed components, which further push the strong baseline to the limit.
>
> [2] Cao C, Ren X, Fu Y. MVSFormer: Multi-View Stereo by Learning Robust Image Features and Temperature-based Depth. TMLR2022.
> [3] Hu E J, Shen Y, Wallis P, et al. Lora: Low-rank adaptation of large language models[J]. arXiv preprint arXiv:2106.09685, 2021.

---

> ### Author Response · Authors · 2023-11-16
> **Response to Reviewer 4iWa (part2/2)**
>
> **4.Validating the proposed modules into other baseline methods (with more pre-trained models or without pre-training).**
>
> Thanks. We should clarify that our work just focuses on transformer usage in MVS rather than pre-training for MVS.
> MVSFormer[2] has thoroughly discussed the effect of different types of pre-training on MVS.
> To further prove the effectiveness of our proposed method, we provide more detailed experiments in the revision in Table13 of Appendix A.5.
>
> In Table13, we add our contributions (CVT and SVA) to DINOv1 pre-training based MVSFormer (MVSFormer-P) and CasMVSNet (without pre-training).
> We re-train CasMVSNet (CasMVSNet*) as an intermediate baseline for a fair comparison, which contains a 4-stage depth hypothesis (32-16-8-4) and cross-entropy loss, sharing the same setting with MVSFormer and MVSFormer++.
> Since the proposed SVA is a side-tuning module specifically designed for pre-trained models, we only evaluate the effect of SVA on MVSFormer-P. From Table.13, our CVT demonstrates substantial improvements for both CasMVSNet* and MVSFormer-P, and our SVA further enhances the results of MVSFormer-P with CVT.
> These results demonstrate the generalization of all proposed components. We added these discussions to the paper revision.
>
> **5.Missing visual comparisons of depth maps of MVSFormer and MVSFormer++.**
>
> Thanks. We added qualitative comparisons and related discussions for the depth map in Figure9 of Appendix, Our MVSformer++ can estimate more precise depth maps even in challenging scenes.
>
> **6.Lacks a discussion of the limitations and failure cases.**
>
> Thanks for this point. We have provided related discussions in Appendix A.8. Though MVSFormer++ enjoys powerful MVS capability as well verified in our experiments, it still suffers from similar limitations as other coarse-to-fine MVS models.
> Specifically, the coarse stage struggles for inevitable error estimations for tiny foregrounds in low-resolution inputs, resulting in error accumulations for the following stages as shown in Figure12. Designing a novel dynamic depth interval selection strategy would be a potential solution to handle this problem, which can be seen as interesting future work.

---

> ### Author Response · Authors · 2023-11-20
> **Response to Reviewer 4iWa (more discussion about fine-tuned DINOv2 and cost-benefit analysis)**
>
> **1.Supplemental result of DINOv2 full-model fine-tuning.**
>
> We apologize for the delay in submitting the full-model fine-tuning result of DINOv2, as it requires a significant amount of computation. As mentioned in MVSFormer[2], we carefully set the learning rate of DINOv2 backbone to 3e-5, while all other learning rates for MVS modules are still 1e-3.
> All results of DINOv2-based MVS are compared as follows:
>
> | Exp      | Acc    | Comp   | Overall |
> |----------|--------|--------|---------|
> | DINOv2 (frozen)   | **0.3198** | 0.2549 | 0.2875  |
> | DINOv2 (LoRA) | 0.3239 | **0.2501** | **0.2870**  |
> | DINOv2 (fine-tuned) | 0.3244 | 0.2566 | 0.2905  |
>
> We would clarify that fine-tuning the whole DINOv2 is costly and achieves inferior performance compared with the frozen one. This phenomenon makes sense. Since DINOv2 is a robust model pre-trained on large-scale curated data, the MVS fine-tuning is based on the limited DTU, which degrades the generalization of DINOv2’s matching capability.
> We added related results and discussion to the Appendix.
> Based on these results, our usage of frozen DINOv2 is convincing.
>
> **2.Cost-benefit analysis of DINOv2. Parameters increasing or module improving?**
>
> Thanks. Since we have accomplished all related experiments of DINOv2, we can prove the improvement does not rely on the cost of more trainable parameters of DINOv2. More importantly, fine-tuning DINOv2 with more trainable parameters could not achieve equivalent improvements as our proposed modules (Overall 0.2905 vs 0.2805).
>
> On the other hand, our proposed SVA is simplified with fewer self-attention blocks, while normalized 2D-PE, FPE (3D-PE), AAS, adaptive layer scaling and Pre-LN introduce substantial improvements for both depth and point clouds with negligible parameters.
> The additional scalability study in Appendix A.2 (Table11) also demonstrates the good efficiency of our model.
> So the superior performance of MVSFormer++ relies on reasonable model designs rather than simply using more parameters.

---

### Official Review · Reviewer_Wn6x · 2023-10-31

**Soundness:** 2 fair
**Presentation:** 2 fair
**Contribution:** 2 fair
**Rating:** 5
**Confidence:** 2

**Summary:**

This paper enhances MVSFormer by infusing cross-view information into the pre-trained DINOv2 model and exploring different attention methods in both feature encoder and cost volume regularization. It also dives into the detailed designs of the transformer in MVS, such as the positional encoding, attention scaling, and position of LayerNorm.

**Strengths:**

1. This paper explores the detailed designs of attention methods in the context of MVSNet.
2. It exploits the pre-trained DINOv2 in the feature encoder and merges the information of source views by cross-view attention.
3. It designs a 3D Frustoconical Positional Encoding on the normalized 3D position, which is interesting and shows good improvements in depth map accuracy.
4. It validates that attention scaling helps the scaling of the transformer to different resolutions, and the position of LayerNorm can affect the final accuracy.

**Weaknesses:**

Although the MVSFormer++ modifies the base model MVSFormer by DINOv2, SVA, Norm& ALS, FPE, etc, the core contributions share similar designs with other MVS methods.

1. In the feature encoder, the Side View Attention is similar to the Intra-attention and Inter-attention in Transmvsnet. The main differences are that this paper uses a pre-trained DINOv2 as input and removes the self-attention for source features.
2. The use of linear attention in the feature encoder has already been proposed in Transmvsnet.
3. In Table 9, although with a larger network, the MVSFormer++ only improves on MVSFormer by a small margin, which can not fully support the claim of the effectiveness of 2D-PE and AAS.
4. The FPE in Table 4 shows good improvement on CVT. The detailed network structure should be made more clear. Please see the questions.
5. The evidence for the minor changes such as the LN and AAS is not strong with experiments on only DTU. They are more intuitive and may need more experiments to prove whether they are generalizable designs. For example, Table 9 on ETH3D actually cannot fully support AAS.

**Questions:**

1. I would to know the detailed structure differences between CVT and CVT+FPE. CVT is only used in the first coarse stage so how many stages use the CVT+FPE in Table 4? What are the results when CVT and CVT+FPE are both used in all stages or only the first coarse stage?
2. The paper can be improved by focusing more on the novel and interesting designs such as the FPE and analyzing more on it.

---

> ### Author Response · Authors · 2023-11-16
> **Response to Reviewer Wn6x (part1/2)**
>
> Thanks for the valuable feedback on our work. We would like to clarify some misunderstandings from the reviewer about the contribution of our work. And we would like to engage in further discussions to address any confusion.
>
> **1.Side View Attention (SVA) is similar to the Intra-attention and Inter-attention in Transmvsnet.**
>
> Thanks for this comment. Besides the pre-trained DINOv2 features and highly simplified architecture (only self-attention for reference features), SVA makes a substantial contribution as a "side-tuning module" [1] based on a frozen pre-trained model (refer to Sec 3.1.1, sentence 2). It's crucial to highlight that no gradients are propagated through the frozen DINOv2+SVA, eliminating the need to store the middle tensor of DINOv2 (achieved with torch.no_grad), resulting in significant GPU memory savings. This way is very efficient to incorporate cross-view information into monocular pre-trained vision models. To the best of our knowledge, we are the first to learn cross-view information for DINOv2 with efficient side-tuning. Moreover, we carefully design the normalized 2D-PE for the robustness of various image scales, the position of layer norm (Norm), and adaptive layer scaling (ALS) for stable and better training convergence, pushing the overall performance to the limit (0.2847->0.2805). As the development of large pre-trained models and transformers for 3D vision, our contributions built upon incrementally learning cross-view information for pre-trained ViTs, and detailed transformer designs for feature encoding are beneficial to the MVS community.
>
> [1] Zhang J O, Sax A, Zamir A, et al. Side-tuning: a baseline for network adaptation via additive side networks. ECCV2020.
>
> **2.Linear attention has been proposed in Transmvsnet.**
>
> Thanks for this point. Although Transmvsnet leveraged linear attention for feature learning, they involved a compromise to save the attention computation, which leaks in-depth discussion about transformer usage in MVS. In our study, we thoroughly investigate the impact of various attention mechanisms for different MVS parts, as outlined in Table 5. As shown in Table5(right), linear attention became the best choice of **feature encoding** because of the global respective fields and image scaling robustness as discussed in Sec.3.1.1 and Sec4.2.3. We have also revealed an insightful finding that **linear attention enjoys good performance in feature encoding, but it performs very terribly in cost volume (Table5 left)**. Since features in cost volume are built upon feature similarity, leaking the essentially informative feature presentation, feature aggregating-based linear attention cannot handle the cost volume regularization compared with spatial aggregating-based attention (discussed in Sec3.2.1 paragraph2). We hope that such an insightful contribution could benefit the community for a better combination of transformer and MVS learning.
>
> **3.Improvements of MVSFormer++ compared to MVSFormer are marginal in Table9.**
>
> Thanks. We need to clarify that our ETH3D results are all achieved with the same threshold (0.5) of depth confidence filter and default settings of dynamic point cloud fusion (DPCD)[2] without any cherry-pick hyper-parameter adjusting. This approach ensures a fair and unbiased evaluation. Though improvements may not be very significant, they are consistent and indicative of the robustness and competitiveness of MVSFormer++.  We have claimed these points in the paper revision.
>
> [2] Yan J, Wei Z, Yi H, et al. Dense hybrid recurrent multi-view stereo net with dynamic consistency checking. ECCV2020.
>
> **4.More experiments to prove the effectiveness of the proposed components, such as Normalization 2D-PE and AAS.**
>
> Thanks for this valuable feedback. We appreciate the reviewer's suggestion for additional experiments to further establish the effect of the proposed components, including Normalized 2D-PE, FPE, and AAS. To further prove the effect of our proposed methods for handling high-resolution scenes, we detail additional ablation studies to Table12 in Appendix A.3 with different image sizes and discuss their influence on MVS depth estimation.
> Both high-resolution depth and point cloud results outperform the low-resolution ones, which proves the importance of adaptability for high-resolution images. Moreover, FPE and AAS improve the results under both 576\*768 and 1152\*1536 images. However, AAS is more effective for depth estimation in high-resolution cases, while the depth gap in low-resolution ones is not obvious. The normalized 2D-PE plays a very important role in high-resolution feature encoding, contributing substantial improvements in both depth and point clouds.
> We have also added these results and discussions to the paper revision.
> For the LayerNorm and adaptive layer scaling, we clarify that these techniques are important to stabilize the SVA training based on frozen DINOv2, which has already been verified clearly in Table4 and Table6.

---

> ### Author Response · Authors · 2023-11-16
> **Response to Reviewer Wn6x (part2/2)**
>
> **5.More details about CVT and FPE; results of CVT+FPE in all stages.**
>
> Thanks for this valuable comment. We have to clarify both CVT and FPE are only used in the first stage (1/8 resolution) as introduced in Sec3.2.1 paragraph2. So CVT+FPE is only applied to the first stage in Table4. As discussed in Sec.3.2.1, we find that CVT for other stages would cause inferior performance. Because only the first stage in a cascade model enjoys a complete and continuous 3D scene. Therefore, the integrality and continuity of the cost volume are very important for CVT. Although FPE is the key to unlocking the capacity of CVT (like position encoding for Transformer), for the 3DCNN-based cost volume regularization of other stages(>1), FPE is not necessary. Because CNN with zero padding could provide sufficient position clues[3]. We further clarify this in the paper revision.
>
> [3] Islam M A, Jia S, Bruce N D B. How much position information do convolutional neural networks encode? ICLR2020.

---

> ### Comment · Reviewer_Wn6x · 2023-11-20
> **Thanks for the response and further questions.**
>
> Thanks for the detailed response from the authors. I have further questions about the rebuttal and the revision.
>
> 1. The SVA and the Intra-attention and Inter-attention in Transmvsnet.
>
> The rebuttal has explained the detailed differences between the SVA and the Intra-attention and Inter-attention in Transmvsnet. The contribution of this part should focus more on introducing the pre-trained ViT into the existing Intra-attention and Inter-attention framework. The paper must be revised to discuss more on the existing Intra-attention and Inter-attention methods instead of referring to the "Side-tuning" method because we need to focus more on the network differences in the context of the practical operations of MVS. With this discussion, this part can make the new contributions such as the DINOv2 more clear.
>
> 2. The linear attention.
>
> The explanation of "in-depth discussion and more experiments" is not very strong because from Table 5, the improvement of linear attention v.s Vanilla is minor according to the $e_2$ and $e_4$. The previous motivation of saving memory makes more sense. The paper revision should also discuss more on the usage of linear attention in existing MVS methods.
>
>
> 3. Improvements of MVSFormer++ compared to MVSFormer are marginal in Table9.
>
> The MVSFormer also used dynamic point cloud fusion (DPCD) for ETH3D. Therefore, this part of the rebuttal does not provide more information about the marginal improvement.
>
> 4. The LayerNorm and adaptive layer scaling.
>
> Thanks for the more experiments on the different image scales. For the LayerNorm and adaptive layer scaling, the results in Table 4 and Table 6 are rather minor: 0.2815 v.s 0.2805 in Table 4, and 0.2827 v.s 0.2850 in Table 6. Without Norm&ALS, the method performs even better on $e_2$ and $e_4$ in Table 4. The depth error is a more direct evaluation metric so I think the strength of Norm&ALS are not clearly supported by Table 4, although this paper discusses a lot about it. From my opinion, since it is not fully supported by the experiments, this paper may degrade this contribution.
>
> Thanks again for the authors' efforts of providing more experiments and discussions.

---

> > ### Author Response · Authors · 2023-11-20
> > **Thanks for the further discussions from Reviewer Wn6x**
> >
> > We appreciate the constructive feedback and active discussion from reviewer Wn6x. We revised our paper with more detailed clarifications about inter-, intra-attention, and linear attention. We also clarified the contribution of Norm&ALS and tuned down the innovative claim of them. Here we provide some more discussion about these questions.
> >
> > **1.The SVA and the Intra-attention and Inter-attention in Transmvsnet.**
> >
> > Thanks for your valuable feedback! We are pleased to reach an agreement with reviewer Wn6x on the contribution of SVA. We further summarize the differences between SVA and Intra, Inter-attention at the last of Sec.3.1.
> >
> > Despite some similarities in using self and cross-attention, our SVA differs significantly from Intra, Inter-attention in both purpose and implementation.
> > The most critical difference is that SVA performs cross-view learning on both DINOv2 (1/32) and coarse MVS (1/8) features (Figure2), while Intra, Inter-attention only considers coarse MVS features.
> > For DINOv2 features, SVA is specifically designed as a side-tuning module without gradient propagation through the frozen DINOv2, which efficiently incorporates cross-view information to this monocular pre-trained ViT.
> > ALS is further proposed to adaptively learn the importance of various DINOv2 layers, while Pre-LN is adopted to improve the training convergence.
> > For coarse MVS features, we emphasize that normalized 2D-PE improves the generalization in high-resolution MVS.
> > We also omit self-attention for source views to simplify the model with competitive performance.
> >
> > Thanks again for this comment, which makes our contribution to SVA clearer in the revision.
> >
> > **2.The explanation of linear attention is not very strong.**
> >
> > Thanks for this comment. We should clarify that we not only prove the good performance of linear attention in feature encoder attention (Table5 right), but also show important evidence that linear attention can not perform well in cost volume learning (Table5 left), which is also a very important opinion to support our contribution of “Tailored attention mechanisms for different MVS modules”.
> >
> > Moreover, we agree with the reviewer Wn6x that both linear and vanilla+AAS attention perform competitively in feature encoder, but linear attention is naturally robust for various image sizes without attention dilution. Thus it can be seen as a more reasonable and efficient choice to be applied for cross-view feature learning in SVA.
> > Generally, the using of linear attention in the feature encoder is complementary to the usage of vanilla attention+AAS+FPE in the cost volume stage (CVT).
> > We have discussed these in Sec.4.2.3 and the last of Sec.3.1.1, and we further complete this opinion in Sec.4.2.3 of the revision.
> >
> > **3.Improvements of MVSFormer++ compared to MVSFormer are marginal in Table9.**
> >
> > Thanks for this comment. Indeed, we also notice that the improvement is relatively marginal if compared to our improvements over other datasets. Nevertheless, we would like to point out that (1) the MVS is indeed a very challenging task; and the MVSFormer is a really very competitive method, which has led the intermediate leaderboard of Tanks and Temple over the past whole year; (2) our method enjoys consistent improvements compared to MVSFormer in various datasets including DTU Dataset, Tanks-and-Temples, and ETH3D. This means the improvement of our method is significant enough, showing the efficacy of our method.
> >
> > **4.The LayerNorm and adaptive layer scaling.**
> >
> > Thanks for this point. We agree with the reviewer Wn6x that Norm&ALS is just a more suitable transformer design to learn cross-view features of DINOv2 (SVA) rather than our main contribution. We have tuned down the innovative claim of them in Sec.3.1.3.
> >
> > However, we have to clarify that depth annotations of DTU are not complete and absolutely correct as discussed in [4]. Thus the results of the point cloud should be more convincing. Moreover, Norm&ALS is mainly working for better feature learning in low-resolution (DINOv2 feature), so it results in fewer large-depth errors. This result conforms to the expectation of DINOv2+SVA which learns more robust low-resolution depth, while precise high-resolution depth should be learned by convolutions. Better results of point clouds also prove our aforementioned opinion.
> >
> > As we further verified in the training log of Fig.13, Appendix A.1, Norm&ALS enjoys faster convergence. Particularly, the best checkpoint of Norm&ALS is achieved in epoch 10, while the best one of MVSFormer++ without Norm&ALS is in epoch 14.
> >
> > Because the figure order has to be maintained for the rebuttal, we are sorry about the disordered figures in the Appendix. And we would re-order these figures in the final version.
> >
> > [4] Luo K, Guan T, Ju L, et al. Attention-aware multi-view stereo CVPR2020.

---

### Official Review · Reviewer_3kfC · 2023-10-31

**Soundness:** 4 excellent
**Presentation:** 4 excellent
**Contribution:** 3 good
**Rating:** 8
**Confidence:** 3

**Summary:**

This paper presents an enhanced iteration of MVSFormer named as MVSFormer++. The method utilizes the Side View Attention (SVA) to empower the cross-view learning ability of DINOv2. It prudently maximizes the inherent characteristics of attention to enhance various components of the MVS pipeline. The results MVSFormer++ achieves on the DTU and Tanks-and-Temples benchmarks show the model works quite well.

**Strengths:**

1. The design of Side View Attention (SVA) is effective.
2. Compared to other models, MVSFormer++ has better performance.
3. The FPE and AAS are used efficiently to generalize high-resolution images.
4. The paper is well written, and one can easily grasp the main idea.

**Weaknesses:**

1. In the ablation study, the results of Norm&ALS under the depth error ratios of 2mm and 4mm are slightly inferior.
2. A discussion regarding the limitations is missing.
3. Minor: Section 4.1 Experimental performance, mean F-score is 41.75 on the Advanced sets in the text while in Tab.3 it is 41.70.

**Questions:**

Please refer to the weaknesses above.

---

> ### Author Response · Authors · 2023-11-16
> **Response to Reviewer 3kfC**
>
> Dear reviewer 3kfC,
>
> Thanks for the valuable feedback on our work. We provide more details about the experiments and limitations. And we would like to engage in further discussions to address any confusion.
>
> **1.Depth results of 2mm and 4mm are inferior with Norm&ALS.**
>
> Thanks for this point. We have to clarify that depth annotations of DTU are not complete and absolutely correct as discussed in [1]. Thus the results of the point cloud should be more convincing. Moreover, Norm&ALS is mainly working for more stable feature learning in low-resolution (DINOv2 feature), so it results in fewer large-depth errors. This result conforms to the expectation of DINOv2+SVA which learns more robust low-resolution depth, while precise high-resolution depth should be learned by convolutions. Better results of point clouds also prove our aforementioned opinion.
>
> [1] Luo K, Guan T, Ju L, et al. Attention-aware multi-view stereo CVPR2020.
>
> **2.Missing limitation discussion.**
>
> Thanks for this point. We have provided related discussions in Appendix A.8. Though MVSFormer++ enjoys powerful MVS capability as well verified in our experiments, it still suffers from similar limitations as other coarse-to-fine MVS models.
> Specifically, the coarse stage struggles for inevitable error estimations for tiny foregrounds, resulting in error accumulations for the following stages as shown in Figure12. Designing a novel dynamic depth interval selection strategy would be a potential solution to handle this problem, which can be seen as interesting future work.
>
> **3.Wrong text description.**
>
> We apologize for the typo here. We revised the result to 41.70 in the revision.

---

### Official Review · Reviewer_B1hX · 2023-10-31

**Soundness:** 3 good
**Presentation:** 3 good
**Contribution:** 3 good
**Rating:** 6
**Confidence:** 4

**Summary:**

This work proposes an enhanced version of MVSFormer. In particular, it specifically addressed three challenges that remained in previous works: tailored attention mechanisms for different MVS modules, incorporating cross-view information into pre-trained ViTs, and enhancing Transformer's length extrapolation capability. Experimental results demonstrated the proposed MVSFormer++ attains state-of-the-art results across multiple benchmark datasets, including DTU, Tanks-and-Temples, BlendedNVS, and ETH3D.

**Strengths:**

+ The contributions of this work are solid and well address the limitations of previous MVS methods. For example, introducing side view attention significantly elevates depth estimation accuracy, resulting in substantially improved MVS results.
+ The combination of frustoconical positional encoding and adaptive attention scaling is interesting. It enhances the model's ability to generalize across a variety of image resolutions while avoiding attention dilution issues.
+ The experiments are comprehensive and promising. Almost all classical and SOTA methods are considered in the comparison experiments, which are evaluated on various datasets. For visual comparisons, the proposed method significantly outperforms other competitive methods, showing more complete structure and fewer geometric distortions.

**Weaknesses:**

- Except for the customized designs beyond the MVSFormer, this work leverages DINOv2 as a new backbone (compared to DINO used in the MVSFormer). It would be interesting to see how the performance of MVSFormer++ changes when it keeps the same backbone as that of MVSFormer.
- MVSFormer and MVSFormer++ show different reconstruction performances regarding different cases on Tanks-and-Temples (Table 3). The authors are suggested to provide more discussions on how the qualitative results differ (like local details and global distributions) and why the degenerations happen.
- The performance of the complete version of this work in the ablation study is different from the quantitative results reported in Table 2. Please elaborate on this inconsistency in metrics.
- The baseline version of MVSFormer (without CVT, FPE, AAS, SVA, Norm&ALS) seems kind of strong already. Does it gain from the strong backbone? Moreover, the qualitative results of the ablation study are expected to be provided.
- The description of Normalization and Adaptive Layer Scaling is ambiguous and unclear. More details about the motivation and implementation would be helpful to understand this part.

**Questions:**

Please refer to the weaknesses.

---

> ### Author Response · Authors · 2023-11-16
> **Response to Reviewer B1hX (part1/2)**
>
> Dear reviewer B1hX,
>
> Thanks for the valuable feedback on our work. We provide more discussions about the backbone selection of MVSFormer++ and other aspects to address your confusion. And we would like to engage in further discussions to address any confusion.
>
> **1.Performance changing of DINOv2 backbone in MVSFormer++; does the improvement gain from a strong backbone?**
>
> Thanks for this comment. The baseline result of MVSFormer enhanced by frozen DINOv2 is listed in the first row of Table4. We further compare the baseline of MVSFormer+DINOv2 with MVSFormer as follows (Table10 of the revision):
>
> | ViT backbone | Frozen backbone | Accuracy$\downarrow$ | Completeness$\downarrow$ | Overall$\downarrow$ |
> |--------------|-----------------|----------------------|--------------------------|------------------------------------|
> | DINO-small   | $\checkmark$    | 0.327                | 0.265                    | 0.296               |
> | DINO-base    | $\checkmark$    | 0.334                | 0.268                    | 0.301               |
> | Twins-small  | $\times$        | 0.327                | 0.251                    | 0.289               |
> | Twins-base   | $\times$        | 0.326                | 0.252                    | 0.289               |
> | DINOv2-base  | $\checkmark$    | **0.3198**     | 0.2549                   | 0.2875              |
> | DINOv2-base  | LoRA (rank=16)  | 0.3239               | **0.2501**          | **0.2870**     |
>
> Compared with DINO and Twins, MVSFormer based on DINOv2 enjoys slightly better results with a frozen ViT backbone. This is strengthened by robust visual features from DINOv2 verified as zero-shot cross-domain feature matching experiments in the DINOv2 paper[1]. We further clarify the motivation for selecting DINOv2 in the last of Appendix A.1. Although MVSFormer+DINOv2 is a strong baseline, we need to clarify that our work is orthogonal to the DINOv2. DINOv2 just improved the overall distance from 0.289 to 0.2875 with less computation (frozen backbone compared with Twins), while we further improved the overall distance from 0.2875 to 0.2805 with our novel designs of SVA, CVT, and image scaling adaption. To verify the effectiveness of our contribution, we further show the improvements of the proposed components based on MVSFormer-P (DINOv1) and CasMVSNet as in Tab.13 of Appendix A.5.
> Our CVT demonstrates substantial improvements for both CasMVSNet* and MVSFormer-P, and our SVA further enhances the results of MVSFormer-P with CVT.
> We added these discussions to the paper revision.
>
> [1] Oquab M, Darcet T, Moutakanni T, et al. Dinov2: Learning robust visual features without supervision[J]. arXiv preprint arXiv:2304.07193, 2023.
>
> **2.Qualitative results of the ablation study**
>
> Thanks for this point. We have compared the qualitative ablation study in Appendix Figure8.  From Figure8, CVT and AAS could effectively eliminate the outliers, while SVA with normalized 2D-PE is also critical for precise point clouds.
>
> **3.Discuss the qualitative results on Tanks-and-Temples.**
>
> Thanks. We have compared more qualitative results between MVSFormer and MVSFormer++  in Figure7 of the Appendix. In general, MVSFormer++ achieves much more precise geometric reconstruction (better precision), while MVSFormer shows more complete results in some scenes, such as the “Recall” subfigure of “Playground”. For the trading-off between precision and recall, MVSFormer++ obviously enjoys a superior balance.
> We added these discussions to the paper revision Appendix A.7.
>
> **4.Different results of the full model in Table2 and ablation study.**
>
> We apologize for the confusing results. Both results of MVSFormer++ in the last row of Table2 and ablations share the same model setting, i.e., our full model. The performance difference is just randomness from two different training results. Because of the server allocation issue, we re-trained the model for the ablation, and subsequently fine-tuned it for additional experiments such as Tanks-and-Temples and ETH3D. The outcomes of them are detailed below. The differences are minimal and can be attributed to the inherent randomness in the training process.
>
> | Exp      | Acc    | Comp   | Overall |
> |----------|--------|--------|---------|
> | Table2(old)   | 0.3105 | 0.2503 | 0.2804  |
> | Ablation(new) | 0.3090 | 0.2521 | 0.2805  |
>
> We have revised the results in Table2, and appreciate your careful reading.

---

> > ### Author Response · Authors · 2023-11-23
> > **Dear Reviewer B1hX**
> >
> > We greatly appreciate reaching a consensus with you!

---

> ### Author Response · Authors · 2023-11-16
> **Response to Reviewer B1hX (part2/2)**
>
> **5.More details about motivation and implementation of Normalization and Adaptive Layer Scaling.**
>
> Thanks for this comment. For the normalization, we have discussed Pre-LN and Post-LN in the related work of "LN and PE in Transformers''. We provide more details about them in Appendix A.1. Post-LN is the regular setting of transformer, which normalizes features after the residual addition, while Pre-LN normalizes them before the attention and Feed Forward Network (FFN).
>
> Post-LN: x=LN(x+attn(x)), x=LN(x+ffn(x)).
>
> Pre-LN: x=x+attn(LN(x)), x=x+ffn(LN(x)).
>
> For the MVS encoding, we found that Pre-LN enjoys more significant gradient updates, especially when being trained for multi-layer attention blocks (as discussed in [2]). In Table6, Pre-LN enjoys superior performance, while we find that Post-LN usually results in slower convergence.
> For Adaptive Layer Scaling (ALS), we have analyzed the high variance issue in Figure6(a) of the Appendix, and multiplied learnable coefficients to different layers' features of frozen DINOv2. Thus our model could adaptively learn the significance of different DINOv2 layers, which stabilize the MVS training.
> We have added all related presentations in the paper revision to make our idea clearer for readers.
>
> [2] Wang Q, Li B, Xiao T, et al. Learning deep transformer models for machine translation. ACL2019.

---

> > ### Comment · Reviewer_B1hX · 2023-11-22
> > **Thanks for the Response**
> >
> > Thanks for the detailed response. It addressed most of my concerns, and I am satisfied with the updated experiment as well as its analysis. Thus, I am inclined to recommend accepting this work.

---

### Author Response · Authors · 2023-11-16
**Response to all reviewers**

Dear Reviewers,

We appreciate the valuable advice and positive feedback from all reviewers, such as “the work is solid”, “experiments are comprehensive and promising”, “paper is well written”, and “one of the best-performing MVS approaches”. We also provided detailed responses and a carefully revised paper to clarify some misunderstandings and unclear presentations.
Some simple tables are provided in the rebuttal system, while other tables, figures, and detailed discussions are provided in our revision.
Modified parts in the revision are in red.
We appreciate the time and effort of all reviewers and ACs in reviewing our submission, and hope our work will receive a fair judgment.

Sincerely, Authors.

---

### Author Response · Authors · 2023-11-23
**Dear reviewers**

Dear reviewers,

As the author-reviewer discussion period is concluding in just a few hours, we would greatly appreciate it if you could confirm that you have reviewed both our rebuttal and the revised version of our work. If there are any further questions or concerns on your part, we are committed to addressing them promptly. Alternatively, if no additional issues arise, we kindly request that you consider adjusting your evaluation accordingly. We deeply value your insightful feedback and eagerly await your response. Thank you.

---

### Meta-Review · Area_Chair_k5Bh · 2023-12-09

**Metareview:**

(a) This paper introduces MVSFormer++, an improved version of the MVSFormer model, addressing key challenges in Multi-View Stereo (MVS) methods. The paper's primary contributions include tailored attention mechanisms, integration of cross-view information into pre-trained Vision Transformers, and other detailed designs of the network. The model demonstrates state-of-the-art performance on several benchmarks, attributed to innovations such as side view attention and frustoconical positional encoding.

(b) Strengths: The paper introduces novel components improving depth estimation. It provides extensive validation across multiple datasets. It proposes the effective use of pre-trained ViTs and tailored attention mechanisms.

(c) Weaknesses: The choice of using DINOv2 as a new backbone initially raised concerns about the fairness of the comparisons. Some proposed components are viewed as incremental improvements; Initial ambiguity in certain method descriptions; Lack of thorough discussion on limitations in the initial submission.

Overall, the paper makes some contributions to the MVS field, with strengths outweighing the addressed weaknesses.

**Justification For Why Not Higher Score:**

1. While the paper introduces significant improvements to the MVSFormer model, some aspects of the contributions were perceived by reviewers as incremental changes, such as the Side View Attention and the usage of DINOv2.

2. There were initial concerns about ambiguities in method descriptions and inconsistencies in results. Although these issues were preliminarily addressed in the authors' response, this suggests a need for further refinement.

**Justification For Why Not Lower Score:**

1. The paper introduces significant improvement over the baseline model.

2. The authors have conducted extensive experiments, demonstrating the model's state-of-the-art performance across several benchmark datasets.

3. The authors provided comprehensive responses to all the concerns raised by the reviewers, effectively addressing the raised issues.

---

### Decision · Program_Chairs · 2024-01-16

Accept (poster)